# Derived Optimal Linear Combination Evapotranspiration (DOLCE): a global gridded synthesis ET estimate

Sanaa Hobeichi[1,2], Gab Abramowitz[1], Jason Evans[1] and Anna Ukkola[2]

[1] Climate Change Research Centre, University of New South Wales, Sydney, NSW 2052, Australia
[2] ARC Centre of Excellence for Climate System Science, University of New South Wales, Sydney, NSW 2052, Australia

*Correspondence to*: Sanaa Hobeichi (s.hobeichi@student.unsw.edu.au)

**Abstract.** Accurate global gridded estimates of evapotranspiration (ET) are key to understanding water and energy budgets,
as well as being required for model evaluation. Several gridded ET products have already been developed which differ in their data requirements, the approaches used to derive them and their estimates, yet it is not clear which provides the most reliable estimates. This paper presents a new global ET dataset and associated uncertainty with monthly temporal resolution for 2000–2009. Six existing gridded ET products are combined using a weighting approach trained by observational datasets from 159 FLUXNET sites. The weighting method is based on a technique that provides an analytically optimal linear combination of
ET products compared to site data, and accounts for both the performance differences and error covariance between the participating ET products. We examine the performance of the weighting approach in several in-sample and out-of-sample tests that confirm that point-based estimates of flux towers provide information at the grid scale of these products. We also provide evidence that the weighted product performs better than its six constituent ET product members in four common metrics. Uncertainty in the ET estimate is derived by rescaling the spread of participating ET products so that their spread
reflects the ability of the weighted mean estimate to match flux tower data. While issues in observational data and any common biases in participating ET datasets are limitations to the success of this approach, future datasets can easily be incorporated and enhance the derived product.

## 1 Introduction

Improving the accuracy and understanding of uncertainties in the spatial and temporal variations of Evapotranspiration (ET)
globally is key to a number of endeavours in climate, hydrological and ecological research. Its estimation is critical to water resource, heatwave and ecosystem stress prediction and it provides constraint on the energy, water and carbon cycles. For these reasons, it is also useful for evaluating the performance of land surface models (LSMs). To identify LSM performance issues and diagnose their probable causes, the observed values of LSM inputs and outputs need to be known with sufficient accuracy and precision. At the site scale, FLUXNET (Baldocchi et al., 2001; Baldocchi, 2008), a global group of tower sites that
measures the exchange of energy, water and carbon between the land surface and the atmosphere, provides direct observations

of ET and most of the drivers and outputs of LSMs with soundly quantifiable uncertainty and sufficient accuracy to make diagnostic evaluation possible. This fact has led to a range of model evaluation, comparison and benchmarking studies using FLUXNET data (e.g. Abramowitz, 2005, 2012; Best et al., 2015; Chen et al., 1997; Stöckli et al., 2008; Wang et al., 2007). Haughton et al. (2016) clearly showed that the flux tower data quality is good enough to provide diagnostic constraint on LSMs

at the site scale. However, the point scale is not the spatial scale at which these models are typically used. In either climate projections or weather forecast applications, gridded model estimates are required at the regional or global scale, with grid cell surface areas of order 25 km$^2$–40 000 km$^2$. The relevance of model diagnostic information at the site scale to broader scale gridded simulations remains unclear.

The advancement of remote sensing technology and satellite image processing techniques has dramatically improved the estimation of many components of LSM forcing (e.g. surface air temperature, precipitation, radiation components and vegetation properties) (Loew et al., 2016). These datasets together with in-situ surface observations have provided constraint on the reanalysis products that provide the basis of global gridded LSM forcing products. On the other hand, very few LSM outputs (such as evapotranspiration or sensible heat flux) can be directly observed by remote sensing, and the only currently

available way to derive them is by the use of modelling schemes or empirical formulations that use satellite based datasets (Jung et al., 2011; Fisher et al., 2008; Miralles et al., 2011; Martens et al., 2016; Mu et al., 2007; 2011; Su, 2002; Zhang et al., 2016; 2010) . This approach leads to large and usually unquantifiable uncertainties. As a result, gridded LSM evaluation products show considerable differences (Ershadi et al., 2014; Jiménez et al., 2011; McCabe et al., 2016 ; Michel et al., 2016; Vinukollu et al., 2011). If an accurate description of these uncertainties was available, then it might be possible to evaluate

gridded LSM simulations in certain circumstances, but until now very few studies quantify LSM uncertainties at regional or global scales (Badgley et al., 2015; Loew et al., 2016; Zhang et al., 2016).

Several gridded ET products have already been developed (see Table 1). These ET products differ in their data requirements, the approaches used to derive them and their estimates (Wang and Dickinson, 2012). So far, it is not at all clear which product provides the most reliable estimates. Recently, inter-comparison studies have aimed to evaluate and compare the available

gridded ET products. For instance, Vinukollu et al. (2011), ran simulations of the Surface Energy Balance System SEBS, Penman–Monteith scheme by Mu (PM-Mu) and modified Priestley–Taylor (PT–JPL) forced by a compiled gridded dataset. The monthly model ET estimates were then compared with ET observations from 12 eddy-covariance towers. The root mean square deviation (RMSD) and biases for all the models fell within a small range with the highest correlation with tower data for PT–JPL, and the lowest for SEBS. These results somewhat disagree with the work of Ershadi et al. (2014) who inspected

the performance of SEBS, the single-source Penman–Monteith, advection-aridity (AA) complementary method (Brutsaert and Stricker, 1979) and PT–JPL using a high quality forcing dataset from 20 FLUXNET stations and found that the models can be ranked from the best to the worst model as: PT–JPL followed closely by SEBS then PM and finally AA. In the same study, a more detailed analysis revealed that no single model was consistently the best across all landscapes. More recently, in the Energy and Water cycle Experiment (GEWEX) LandFlux project, McCabe et al. (2016) ran simulations of the algorithms in

SEBS, PT–JPL, PM-Mu and Global Land Evaporation Amsterdam Model (GLEAM) from common global-scale gridded forcing data, as well as site-based forcing data, and assessed their response relative to ET measurements from forty-five globally distributed FLUXNET towers. The results indicated that PT–JPL achieved the highest statistical performance, followed closely by GLEAM, whereas PM-MU and SEBS tended to under- and over- estimate fluxes respectively.

Furthermore, in the Water Cycle Multi-mission Observation Strategy for Evapotranspiration (WACMOS-ET) project-part 1, Michel et al. (2016) carried out validation experiments for PT–JPL, PM-MU, SEBS and GLEAM against in situ measurements from 85 FLUXNET towers and found that there was no single best performing model.

In each of the evaluation studies described above, tower data from FLUXNET provide ground truth for gridded ET datasets by comparing grid cell values to those measured at the site scale. Most gridded ET products have a 0.5-degree resolution, so

that each grid cell can represent an area of around 2500 km$^2$. The fetch of flux tower measurements varies depending on terrain, vegetation and weather, but is typically under 1 km$^2$ (Burba and Anderson, 2010) . None of these studies directly address this obvious scale mismatch, and the degree to which surface heterogeneity might nullify any information that flux towers provide about fluxes at these larger scales. Indeed, all the evaluation studies that compared gridded estimates against flux tower observations highlighted this point as a limitation in the evaluation approach.

Zhang et al. (2016) pointed out that merging an ensemble of gridded ET products using a sophisticated data fusion method is likely to generate a better ET product with reduced uncertainty. Ershadi et al. (2014) and McCabe et al. (2016) noted in their studies that the multi-product mean produces improved estimates relative to individual ET products. Similarly, the analysis result of the LandFlux-EVAL project (Mueller et al., 2013) suggested that deriving ET values using the mean of multiple datasets outperforms the ET values from individual datasets. These results do suggest that flux towers likely provide some

information at the gridded scale, since there are solid theoretical reasons to expect the mean to outperform individual estimates (Annan and Hargreaves, 2010; Bishop and Abramowitz, 2013), although this fact was not noted in these studies. That is, we would expect that the mean of a range of relatively independent approaches – whose errors are somewhat uncorrelated – would provide a better estimate to observations if those observations were of the same system being simulated. In this study, we provide an even stronger vindication of this relationship in section 2.3 below.

This paper introduces a new method for deriving globally gridded ET as well as its spatio-temporal uncertainty by combining existing gridded ET estimates. The method is based on the ensemble weighting and rescaling technique suggested by Bishop and Abramowitz (2013). The technique provides an analytically optimal linear combination of ensemble members that minimizes mean square error when compared to an observational dataset, and so accounts for both the performance differences and error covariance between the participating ET products (for example, caused by the fact that different ET datasets might

share a forcing product). In this way, at least in-sample, the optimal linear combination is insensitive to the addition of redundant information. We use a broad collection of FLUXNET sites as the observational constraint.

We examine the performance of the weighting approach in several in-sample and out-of-sample tests that confirm flux towers do indeed provide information at the grid scale of these products. Having confirmed this, we note that this "Derived Optimal

Linear Combination Evapotranspiration" (DOLCE) is more observationally constrained than the individual participating gridded ET estimates and therefore provides a valuable addition to currently available gridded ET estimates.

Sections 2.1 and 2.2 present the gridded ET datasets and the flux tower ET used to derive the weighted ET product. Sect. 2.3 describes the weighting approach used to calculate the linear combination ET and its uncertainty. In Sect. 2.4 we explain our methods for testing the weighting approach and the representativeness of the point scale to the grid scale. We then present and discuss our results in Sect. 3 and 4 before concluding.

## 2 Methods, data and experimental setup

To build the DOLCE product, six global gridded ET products, all classified as diagnostic datasets based on Mueller et al. (2011), are weighted based on their ability to match site-level data from 159 globally distributed flux tower sites. DOLCE is derived at 0.5° spatial resolution and monthly temporal resolution for 2000–2009.

## 2.1 ET datasets

This study employs monthly values from ten existing gridded ET datasets (Table 1). Only six of these datasets – referred to in this study as the Diagnostic Ensemble – are used for building DOLCE, while all of the ten are used to evaluate its performance (we'll refer to these as the Reference Ensemble). The reasons for restricting the Diagnostic Ensemble are (a) to maximize the time period covered by DOLCE (see Table 1), (b) to avoid temporal discontinuities in the derived product that can result from using different component products in different time periods ,(c) to maximize the number of flux tower sites that can inform the weights (noting that datasets have different spatial coverage), and (d) to avoid LSM-based estimates in the final DOLCE product, so that its validity for LSM evaluation is clearer. For these reasons, CSIRO-global, LandFlux-EVAL-ALL, LandFlux-EVAL-DIAG and PT–JPL were removed from the Diagnostic Ensemble.

The weighting approach becomes increasingly effective as more products are included in the weighted mean, but since different products have different spatial coverage, there is a need to strike a balance between maximizing the number of products included in DOLCE and maximizing spatial coverage of DOLCE. To resolve this issue, three different subsets of the Diagnostic Ensemble were chosen to derive DOLCE "tiers". The three DOLCE tiers differ in their spatial coverage of ET and uncertainty, the number of ET products included, and the number of flux towers employed to compute them. DOLCE tier 1 was derived by employing all the products in the Diagnostic Ensemble and 138 sites, and has a spatial coverage equal to the intersection of the global land covered by these products, as shown in Red in Fig. 1. In DOLCE tier 2, GLEAM v2B was excluded and the remaining products were used to compute ET and uncertainty, increasing the spatial coverage and allowing 151 observational flux tower sites. The newly added regions in DOLCE tier 2 are shown in orange in Fig. 1. Finally, ET and uncertainties in the regions shown in yellow were computed using only two ET products, GLEAM V2A and GLEAM V3A, to create DOLCE tier 3, using 159 flux sites. With this approach, we manage to create a global product without sacrificing data quality in regions where more information is available.

## 2.2 Flux tower data

The flux tower data used in this work is a composite of daily values from the FLUXNET2015-Tier 1 (FN) and LaThuile2007 (LT) Free Fair Use databases (Baldocchi, 2008; Baldocchi et al., 2001; Papale et al., 2012; ORNL DAAC, 2015). We applied quality control and filtering to the site data as follows:

5   1.      Omit the observations that don't fall within the temporal coverage of DOLCE.

2.      Omit the daily ET observations if less than 50% of half-hourly ET was observed on that day (as opposed to gap-filled).

3.      Omit the monthly ET aggregates that have been calculated from less than 15 daily mean ET values.

4.      Correct the remaining site-months of ET observations from the LT dataset for energy balance non-closure on a site-by-site basis (energy closure corrected FN daily data was used)

5.      Keep only the sites that are located within the spatial coverage of products in any of the three tiers of the Diagnostic Ensemble (Fig. 2).

6.      Exclude irrigated sites or those that have only one monthly record.

Applying (1), (2) and (3) reduced the number of tower sites that were available at the beginning of the analysis from 246 to 172 sites, and produced a total of 7891 site-months (out of a possible 18468). Applying (5) and (6) reduced the number of sites against which the weighting is tested to 138 sites in DOLCE tier 1, increasing to 151 sites in DOLCE tier 2 and 159 in DOLCE tier 3. In (4), two different correction techniques were applied for energy balance non-closure at LT sites. Both involve ensuring that $R_n - G = H + LE$ where $R_n$ is the net radiation, G, H and LE are the ground heat flux, sensible heat flux and latent heat flux respectively, at either the monthly timescale or over the length of the entire site record. Applying a correction technique for energy imbalance at LT sites required applying (2) and (3) for the other components of energy imbalance (i.e. $R_n$, G and H), which means that the sites that had to undergo a correction for the energy imbalance, should have monthly estimates for all the fluxes of the energy budgets, where each monthly value has been calculated from at least 15 daily mean flux values. Because of this constraint, many sites were disregarded from the analysis. One of the two correction method for energy imbalance at LT sites distributes the residual errors in heat fluxes according to the Bowen ratio (BR) (Bowen, 1926; Sumner and Jacobs, 2005; Twine et al., 2000) such that:

$$BR = \frac{H_{uncorrected}}{LE_{uncorrected}} \tag{1}$$

is applied using

$$F_{corrected} = \frac{R_n - G}{H_{uncorrected} + LE_{uncorrected}} \times F_{uncorrected} \tag{2}$$

where F is either H or LE, so that BR is unchanged between corrected and uncorrected fluxes. At first, the Bowen Ratio technique was applied on monthly basis, but this occasionally seemed to give erratic results in the corrected flux values. However, when a single correction per site was applied across all the monthly records, the results were more consistent.

The second approach calculates ET as the Residual term in the Energy Balance Equation $LE_{corrected} = R_n - H - G$. The two correction methods were tested separately but no qualitative differences were noticed. All results below use the BR technique. In (6), we expect that some of the weighting models will largely underestimate the flux at irrigated sites, a result of a missing irrigation module in their scheme (Jung et al., 2011; Miralles et al., 2011). Because of this, the error bias of these models at

the irrigated sites will modify the mean error bias (i.e. mean bias across all the sites) significantly, which will affect the weighting in favour of the products that can represent better irrigation. We excluded these sites as we do not want the products to be weighted for their inclusion/non-inclusion of physical processes.

We used daily averages of latent heat flux represented by "LE_CORR" in FN dataset. In LT dataset, we employed the components of energy imbalance and their associated flags (in brackets), represented by G_f (G_fqcOK), for soil heat flux,

H_f (HFqcOK) for sensible heat flux, Rn_f (Rn_fqcOK) for surface net radiation and LE_f (LE_fqcOK) for latent heat flux.

## 2.3 Weighting approach

The weighting approach presented here was suggested by (Bishop and Abramowitz, 2013). Given an ensemble of ET estimates and a corresponding observational dataset across time and space, the weighting builds a linear combination of the ensemble members that minimizes the mean square difference (MSD) with respect to the observational dataset such that: If $x_k^j$ is the $j^{th}$

time-space step of the $k^{th}$ bias-corrected ET product (i.e. after subtracting the mean error from the product), and a linear combination of the ET estimates is expressed as

$$\mu_e^j = w^T x^j = \sum_{k=1}^{K} w_k x_k^j \quad , \tag{3}$$

then the weights $w^T$ provide an analytical solution to the minimization of

$$\sum_{j=1}^{J} (\mu_e^j - y^j)^2 \tag{4}$$

subject to the constraint that $\sum_{k=1}^{K} w_k = 1$, where $j \in [1, J]$ are the time-space steps, $k \in [1, K]$ represent the ET products, $y^j$ is the $j^{th}$ observed time-space step.

The solution is expressed as

$$w = \frac{A^{-1} 1}{1^T A^{-1} 1}, \tag{5}$$

$$\overbrace{}^{k \text{ elements}}$$
where $1^T = [1, 1, \ldots, 1]$ and A is the k × k error covariance matrix of the gridded products. Further details are in (Abramowitz

et al., 2015; Bishop and Abramowitz, 2013; Zeller and Hehn, 1996).

As noted above, this weighting approach has two key advantages. 1) It provides an optimal solution to minimizing mean square error differences between the weighted ET estimates and the observational tower data in-sample. 2) It accounts for the error covariance between the participating datasets (e.g. caused by the fact that single datasets may share ET schemes or forcing), that is, they may not provide independent estimates. The analytical solution guarantees that the weighted mean will perform

as well or better than the unweighted mean or any individual estimate included in the linear combination, at least on the data used to train the weights (that is, in-sample). Moreover, the addition of a new gridded ET product to the ensemble will not

degrade the performance of the weighted product, even if it consists of a duplicate product or a poor performing product. In order to confirm this in-sample weighting improvement, we performed an in-sample test presented in the supplementary material (Fig. S1). However, there is no guarantee that the weighted product will necessarily perform well out-of-sample, at a collection of sites not used for training. We explore this more below.

We use the ensemble dependence transformation process presented in Bishop and Abramowitz (2013) to calculate the spatiotemporal uncertainty of DOLCE. This involves first quantifying the discrepancy between our weighted ET estimate, $\mu_e$, and the flux tower data, expressed as an error variance, $s_e^2$, over time and space. We then transform our Diagnostic Ensemble so that its variance about $\mu_e$ at a given time-space step, $\sigma_e^{2j}$, averaged over all time and space steps where we have flux tower

data, is equal to $s_e^2$. This process ensures that the spread of the transformed Diagnostic Ensemble provides a better uncertainty estimate than simply using the spread of all products in the original Diagnostic Ensemble, since the spread of the transformed ensemble now accurately reflects uncertainty in those grid cells where flux tower data is available.

We calculate the discrepancy between observations and our weighted mean as

$$s_e^2 = \frac{\sum_{j=1}^{J}(\mu_e^j - y^j)^2}{J-1} \tag{6}$$

Next, we wish to ensure that

$$\frac{1}{J}\sum_{j=1}^{J}\sigma_e^{2j} = s_e^2 \tag{7}$$

but the variance of our existing Diagnostic Ensemble will not, in general, satisfy this equation. To transform it so that it does, we first modify the coefficients from Eq. (3), so that they're guaranteed to all be positive:

$$\widetilde{w} = \frac{w^T + (\alpha-1)\frac{1^T}{K}}{\alpha} \tag{8}$$

where $\alpha = 1 - K\min(w_k)$ and $\min(w_k)$ is the smallest negative weight (and $\alpha$ is set 1 if all $w_k$ are non-negative). We then transform the ensemble using

$$\tilde{x}_k^j = \mu_e^j + \beta(\bar{x}^j + \alpha(x_k^j - \bar{x}^j) - \mu_e^j) \tag{9}$$

where

$$\beta = \sqrt{\frac{s_e^2}{\frac{1}{J}\sum_{j=1}^{J}\sum_{k=1}^{K}\widetilde{w}_k(\bar{x}^j + \alpha(x_k^j - \bar{x}^j) - \mu_e^j)^2}} \tag{10}$$

If we then define the weighted variance estimate

$$\sigma_e^{2j} = \sum_{k=1}^{K}\widetilde{w}_k(\tilde{x}_k^j - \mu_e^j)^2 \tag{11}$$

we ensure that both Eq. (6) above holds, and also that $\mu_e^j = \sum_{k=1}^{K}w_k x_k^j = \sum_{k=1}^{K}\widetilde{w}_k\tilde{x}_k^j$ (see Bishop and Abramowitz, 2013, for proofs). We then use $\sqrt{\sigma_e^{2j}}$ as the spatially and temporally varying estimate of uncertainty standard deviation, which we will refer to below as uncertainty.

**2.4 Experimental setup**

We employed four statistical metrics that reflect how well a gridded ET product represents the quality controlled flux tower observations: Mean Square Error $MSE=mean(\,dataset - observation\,)^2$; Mean Bias$=mean|\,dataset - observation\,|$; Correlation $COR=corr(observation, dataset)$ and Modified Relative Standard Deviation $MRSD = \dfrac{\sigma_{dataset\ or\ observation}}{max(mean(observation),\ q)}$.

We use a modified relative standard deviation metric MRSD that measures the variability of latent heat flux relative to the mean of the flux measured at each site. This ensures that a comparison between MRSD for a product and observations can tell us whether a product's variability is too large or too small (unlike relative standard deviation). The term 'q' is a threshold representing the $2^{nd}$ percentile of the distribution of observed mean flux (i.e. temporal mean ET) across all sites ($\approx 13$ W/m$^2$), which guarantees that MRSD calculated across many sites is not dominated by sites where the mean flux (denominator

in MRSD Equation above) approaches zero. We looked at the bias in MRSD for each product considered- i.e. $|MRSD_{dataset}$ - $MRSD_{observation}|$, and showed the performance improvement of the weighted mean.

In every test, we split the available sites into in-sample and out-of-sample sites (that is, calibration and validation sites). First, we applied the weighting process at the in-sample sites to weight the individual datasets of the Diagnostic Ensemble and derive a weighted product. Then, we calculated the four metrics above using the out-of-sample sites and displayed results by showing:

a.        The percentage performance improvement of the derived weighted product compared to the equally weighted mean (Dmean) of the Diagnostic Ensemble;

b.        The percentage performance improvement of the derived weighted product compared to each individual product in the Reference Ensemble, yielding 10 different values of performance improvement.

c.        The aggregate (Ragg) of the values of performance improvement calculated across all 10 products in the Reference

Ensemble.

We display the results of performance improvement datasets calculated in (a-c) above in 12 box and whisker plots. In each boxplot, the lower and upper hinges represent the first ($Q_1$) and third ($Q_3$) quartiles respectively of the performance improvement datasets and the line located inside the boxplot represents the median value. The extreme of the lower whisker represents the minimum of 1) max(dataset) and 2) ($Q_3 + IQR$), while the extreme of the upper whisker is the maximum of 1)

min(dataset) and 2) ($Q_3 + IQR$)), where IQR is the interquartile range of the performance improvement dataset. If the median performance improvement is positive, this indicates that the weighting offers an improvement in more than half of the data presented in the boxplot.

We first divide sites between the in-sample and out-of-sample groups by randomly selecting 25 % of the sites as out-of-sample. The remaining sites form the in-sample training set, and are used to calculate a scalar bias correction term and a weight for

each participating gridded ET product. These bias correction terms and the weights are then applied to the products at the out-of-sample sites. The test was repeated 5000 times with different random selection of sites being out of sample.

Next, we repeat the process with just one site in the out-of-sample testing group. The bias corrections and weights are therefore derived on all sites except one, and tested on the single out-of-sample site. The same test was repeated for all the participating sites.

## 3. Results

The results for the 25 % out-of-sample test are displayed in the box and whisker plots presented in Fig. 3 (a), (b), (c) and (d). The MSE plot in Fig. 3 (a) indicates that across most of the random selections of the 25 % out-of-sample sites (57 % of random selections), the MSE of the weighted product is slightly better (with maximum 12 % improvement) than the equally weighted mean of the Diagnostic Ensemble. The weighting also improved results for almost all random selections of sites when compared to each individual dataset in the Reference Ensemble. The MRSD plot in Fig. 3 (b) shows that the weighting

succeeded in improving MRSD relative to Dmean and PT-JPL by no more than 4 % and 36 % respectively across the majority of the combinations of sites, but didn't improve MRSD relative to the other reference products. This is perhaps unsurprising, given that both the weighted and unweighted mean should have decreased variability when the variations in individual products are not temporally coincident. The COR plot in Fig. 3 (c) shows that despite the expected drop in variability in the weighted product, the correlation with site data has in fact improved across most of the different random selections of the 25 % out-of-

sample sites relative to all of the Reference Ensemble datasets. The improvement was minimal relative to Dmean (maximum 2 %) and MPI (maximum 3 %). Similarly, the Mean Bias plot (Fig. 3 (d)) shows a clear improvement relative to Dmean and the Reference Ensemble datasets across most combinations of sites. It is important to reinforce that these results are for sites that were not used to train the weights. As detailed in section 2.3, performance improvement at training sites is expected, but the fact that the weighting delivers improvements at sites that were not included in training data indicates that there is indeed

information content about the larger scales in site data.

The results of the one site out-of-sample tests are displayed in Fig. 4 (a), (b), (c) and (d). These boxplots indicate that across three metrics (MSE, MRSD and Mean Bias) the weighted mean clearly outperformed all the products in the Reference Ensemble. However, when the performance improvement of the weighted product is compared against Dmean the improvement was marginal. In MSE, an improvement was achieved at more than half (53 %) of the sites (Fig. 4 (a)). In

MRSD, 51 % of the sites showed a small improvement relative to Dmean (Fig. 4 (b)). In COR, the performance decreased at 57 % of the sites, and a minimal improvement over the Reference Ensemble was shown at more than half of sites (Fig. 4 (c)). Finally, in Mean Bias, 58 % of the sites showed performance improvement over Dmean (Fig. 4 (d)). Part of the success of the weighting approach relative to the multi-product mean is due to the bias correction applied before the weighting. Figure S3 in supplementary material separates the effect of each step.

The results of the one site out-of-sample test suggest that spatial heterogeneity - the discrepancy between site and grid scale estimates - is significant, as the spread of these boxplots is large. These sites are more likely to be where the fetch of the flux

site observations cannot represent the data of the entire grid (estimates). A further analysis (not shown here) has indicated that sites that show poor improvement in Fig. 4 (a), (b), (c) and (d) have a consistently high bias against all of the gridded products. To investigate the effect this spatial heterogeneity is having on the out-of-sample tests, we also repeat the one site out-of-sample experiments using a subset of the sites within grid cells that are deemed to be relatively homogeneous (Fig. 4 (e), (f),

(g) and (h)). Homogeneity is defined in this case by using only those sites that have the same IGBP vegetation type as the grid cell that contains them. Almost all the gridded ET products use the dominant land cover type for computing ET, yet this is not always the same as the land cover type surrounding the flux site. Since this mismatch occurs in heterogeneous terrains, we will denote the sites that have this property as "HET-case" sites, whereas the sites that show agreement with the underlying grid cell vegetation type are denoted as "HOM-case" sites. We retrieve the IGBP vegetation cover data of the grid cells from

MODIS Land Cover for year 2009 at 0.5° spatial resolution (http://glcf.umd.edu/data/lc/; Friedl et al., 2010). The vegetation type of individual sites is taken from metadata on the FLUXNET website. We show the distribution of HET-case and HOM-case sites by biome type in Table S2 in the supplements. There is some expectation that the weighting will show better performance if it is trained with HOM-case sites only, although HOM-case sites consist of about one-thirds of the total number of sites used in this study. To further investigate this idea, we carried out the one site out-of- sample test training and testing

on HOM-case sites data only. Figures 4 (e), (f), (g) and (h) show that the HOM-case subset of tower sites does indeed improve the result marginally at least in terms of MSE, COR and Mean Bias when compared to the original dataset (HOM & HET -case; Fig. 4 (a), (b), (c) and (d)). We did not investigate the 25 % out-of-sample tests with HOM-case sites, since the number of in-sample sites would become too small, and likely lead to overfitting. In both cases it is important to note that many individual sites agree poorly with the weighted product compared to some other products. The distinction between the results

shown in Fig. 3 versus Fig. 4 serves to highlight that DOLCE, and indeed any other large scale gridded ET product, is not suitable for estimation of an individual site's fluxes, even if prediction over many sites shows notable improvement.

On the basis of the aggregate out-of-sample improvement that this approach offers over existing gridded ET products, in terms of MSE, MRSD, COR and Mean Bias against site data, we now present details of the DOLCE product, which is trained using all site data and derived from the combination of the six diagnostic products. The 3 GLEAM products were resampled to 0.5°

using bilinear interpolation to match the spatial resolution of DOLCE.

We calculated four statistics - Mean Bias, RMSE, Standard Deviation (SD) difference (i.e. $\sigma_{DOLCE} - \sigma_{observation}$) and Correlation - to see how well DOLCE performs at each site. The results are displayed in Fig. 5 (a), (b), (c) and (d). At about half of sites, DOLCE had values between –6 and 6 W m$^{-2}$, –5 and 5 W m$^{-2}$ in the Mean bias and SD difference metrics respectively, and values greater than 0.93 W m$^{-2}$ and less than 14 W m$^{-2}$ in correlation and RMSE respectively. DOLCE metrics

vary in the range [–43.4–30.1 W m$^{-2}$ ] for mean bias, [–25.8–23.5 W m$^{-2}$ ] for SD difference, [0.09–0.99 W m$^{-2}$] for correlation and [4.2–60.9 W m$^{-2}$] for RMSE.

We also calculated the on-site metrics (i.e., RMSE, Mean Bias, SD difference, and Correlation) for DOLCE separately at each subset of sites (i.e. HOM-case and HET-case) and we displayed the results in two boxplots (Fig. 5 (left column)). These

boxplots show that most of the sites at which DOLCE showed low performance are HET sites, since the end of whiskers is larger for the HET-case sites.

We tested the performance of DOLCE at three irrigated sites that were excluded from the weighting for reasons explained earlier (section 2.2), by computing the four statistics. A description of these sites and the results are shown in Table 2. The results show that the performance of DOLCE is reasonable at US-Ne1 and US-Ne2 and low at US-Twt. These results are discussed further below.

We now look at the differences between DOLCE and two widely used ET products. Figures 6 and 7 show the seasonal mean difference ET between MPIBGC and DOLCE and between LandFlux-EVAL-DIAG and DOLCE respectively. The differences were computed from seasonal means using the same spatial mask over the period 2000–2009 for MPIBGC and 2000–2005 for LandFlux-EVAL-DIAG. The seasonal plots in Fig. 6 show that, overall, DOLCE has lower ET than MPIBGC. DOLCE tends to have higher ET values in the Asian and Australian tropics during the austral summer and autumn. DOLCE has higher ET values in Sahel during Sep–Nov and in the high plateau of Madagascar during Dec–May. In the Amazon, DOLCE exhibits higher ET values between June and November. Higher values of DOLCE are seen in Guiana highlands throughout the year. The Brazilian highlands show higher values for MPIBGC from June to November. The Mid latitudes of the Northern hemisphere show higher values for MPIBGC between March and August but no significant differences are seen in this area between September and February.

In Fig. 7, there are large differences in ET showing higher values in LandFlux-EVAL-DIAG ET over the Amazon, the rainforests of Southeast Asia and in the high plateau of Madagascar throughout the year. DOLCE tends to exhibit higher ET values in the Ethiopian highlands and Myanmar between June and November.

The spatial distribution of DOLCE mean ET and its seasonal variability (standard deviation) over the austral Summer (Dec–Feb) and Winter (Jun–Aug) from 2000 to 2009 is shown in Fig. 8 (a) and (b) respectively. The seasonal variability of ET is larger in the warm season but is always small over Antarctica, Greenland and the deserts in North Africa (Sahara), the middle east (Arabian Peninsula desert) and Asia (i.e. Gobi, Takla Makan and Thar). The average uncertainty shown in if Fig. 8 (c) is bigger in the warm season, this is in agreement with the relatively large size of the flux in the warm season, and its seasonal variability shown in Fig. 8 (d) is also in agreement with the seasonal variability of the flux. Figure 8 (e) is intended to give some indication of the reliability of DOLCE. Regions in blue show grid cells that satisfy $\frac{\text{Uncertainty SD}}{\text{mean ET}} \leq 1$. Those shown in green have |mean ET|<5 and Uncertainty SD < 10 but don't satisfy $\frac{\text{Uncertainty SD}}{\text{mean ET}} \leq 1$, so that the uncertainty estimates are higher than their associated ET, but both the uncertainty and the ET estimates are very small. All the remaining grid cells, perhaps those least reliable, are shown in red. The global maps in Fig. 8 (e) shows that most of the non-reliable uncertainty values are located in the land added in tier 3 (Fig. 2 in yellow). This is not surprising, because uncertainty values have been derived using only two gridded ET datasets that are also not observationally constrained due to the lack of the sites representing the terrestrial ecosystems and environmental conditions in these areas. Some regions show unreliable uncertainty estimates

during the cold season only, located in the plateau of Tibet, over the Andes, the Australian deserts, the South-African deserts, American Great basin and Rocky Mountains.

## 4. Discussion

Many studies have analysed the systematic and random errors of latent heat flux in FLUXNET measurements (Dirmeyer et al., 2016; Göckede et al., 2008; Richardson et al., 2006). These studies have detected errors of magnitudes that cannot be neglected. A recent study (Cheng et al., 2017) showed that the computed eddy-covariance fluxes have errors in the applied turbulence theory that lead to the underestimation of fluxes, and that this is likely to be one of the causes of the lack of surface energy closure. In this study, we 1) used the flag assigned to the observed flux, to filter out the low-quality data and 2) used energy-balance-corrected FLUXNET data which has higher per-site mean values than the raw data at most of the sites (85% of them). We expect that filtering together with the use of corrected data will reduce the magnitude of the uncertainty in the observational data used here and compensate to a certain extent for the underestimation due to the systematic errors. However, we have not formally explored a range of approaches to addressing this. The possibility of systematic biases in FLUXNET data remains, and this could clearly lead to systematic biases in DOLCE.

We have also assumed that error across sites is uncorrelated, which, given the distribution of sites, is unlikely to be true, meaning that the effective number of sites is probably somewhat smaller than those shown in Fig. 2. Given this dependence is likely to vary depending on a range of time varying factors, we have left the job of attempting to disentangle this issue for future work.

Nevertheless, the results of the one site out-of-sample and 25 % out-of-sample tests above suggest that the weighting of gridded ET products in this way produces a more capable ET product overall. Critically, the fact that the weighting improves out of sample performance suggests that while the representativeness of point-scale measurement for the grid scale may not exist at every single site, it does exist across all these sites as a whole. We also investigated using bilinear interpolation instead of direct grid cell to tower comparison (not shown), but found no qualitative differences.

It might be reasonable to assume that eliminating sites that poorly represent grid cell properties might further improve these results, yet the results of categorizing sites into HOM-case and HET-case subsets suggested that this is not necessarily a simple process. Our characterisation was based on a relatively simple vegetation characterisation, yet there are a range of soil-vegetation interactions that likely affect ET in each location, that are not captured in this classification. Among HET-case sites, for example, the worst performance occurred at AU-Fog. This poor performance is a reflection of the large discrepancy between the land cover seen by the flux tower which is a wetland landscape in a monsoonal climate, and thus characterised by high evaporation rates throughout the year and the Savannah land cover seen by the satellite products where the vegetation dries or dies in the dry season. Also, we note that none of the six diagnostic products was able to reproduce the flux measured at this site. Perhaps unsurprisingly, DOLCE scored the worst values for correlation, RMSE and Mean Bias in AU-Fog.

Aside from AU-Fog, three sites showed low correlation (less than 0.5) between DOLCE and observations. These include ID-Pag, ES-LgS and IL-Yat (see Supplementary Table S1). IL-Yat site is an Evergreen Needleleaf Forest site located over a desert (Bare) grid cell and therefore belongs to the HET-case which might be the reason for the discrepancy between the satellite derived ET and the observed ET. ID-Pag belongs to the HOM-case, but a burning event occurred in a nearby forest between mid-August and late October 2002, and caused the site to be covered with a dense smoke-haze. Although these three months were excluded from the analysis, it is very likely that the burned region produced extremely different ET compared to the surrounding unburned landscape, making it a HET-case site until the regeneration of the vegetation and might be the reason why the weighting products couldn't reproduce ET in the following couple of months. ES-LgS is an Open Shrubland site that belongs to the HOM-case. This site operated from 2007 to 2009 and is currently inactive. The reason of the low correlation of DOLCE with the observed fluxes is not clear, however previous studies (Ershadi et al., 2014; McCabe et al., 2016) stated that the complexity of Shrubland landscape is the cause of the poor performance of many models (PM, GLEAM and PT–JPL). In the case of DOLCE, the poor performance was revealed in one metric only (i.e. correlation). These results nevertheless lead us to expect that if we construct DOLCE by incorporating HOM-case sites only, we might get a better product, but the small number of sites satisfying this property, the fact that the separation of sites into HOM-case and HET-case can lead to a separation of land covers, and the difficulty in defining a meaningful definition for expected flux homogeneity are limiting factors. Determining whether DOLCE performs better at HOM-case sites or HET-case sites is inconclusive. Even that the worst performance of DOLCE was achieved in HET-case sites, the boxplots in Fig. 5 (a), (d) show that the value of the median, lower and upper quartiles are better in HET-case for two metrics (i.e. RMSE and COR). While we expect that calibrating the weighting with HOM-case could lead to a better product, we don't expect to see DOLCE overall performing better in any of the two groups.

Figure 9 shows that DOLCE performs differently for different vegetation types, at least as sampled by the flux towers we use here. Previous studies have shown that different ET estimation approaches perform differently over different biomes. For example, Ershadi et al. (2014) found that the PM models outperform PT–JPL across Grasslands, Croplands and Deciduous Broadleaf Forests, but PT–JPL has higher performance than PM models over Shrublands and Evergreen Needleleaf Forest biomes. McCabe et al. (2016) observed poor performance of GLEAM, PM models and PT–JPL over Shrublands and low performance over the forest biomes and higher performance over short canopies. The number of sites for each vegetation type in our case varies, with four for Open Shrublands (OSH), 39 for Evergreen Needleleaf Forest (ENF), nine for Evergreen Broadleaf Forest (EBF), 18 for Deciduous Broadleaf Forest (DBF), 10 for Mixed Forests (MF), four for Woody Savannas (WSA), four for Savannas (SAV), 42 for grassland (GRA), eight for Wetland (WET), one for Cropland/Natural Vegetation Mosaic (VEG) and 19 for Croplands (CRO). Of the 159 sites that are used to weight the tier 3 products (GLEAM v2A and GLEAM v3A), only 142 sites are within the spatial coverage of DOLCE. This decrease is caused by the coarser resolution of DOLCE compared to GLEAM products. The remaining 17 sites were located on grids that were identified as land grid cells in GLEAM's spatial resolution (0.25°) but water body in DOLCE's spatial resolution (0.5°). The EBF box and whisker plot in Fig. 9 (d) shows the correlation of DOLCE at eight EBF sites, out of which two sites are located in the tropics. The lowest

correlation seen in this biome type is at the tropical sites ID-Pag (0.26) and BR-Sa3 (0.62). This suggests that DOLCE tends to represent ET at the extratropical sites better than the tropics, and this is not surprising since most of the sites that were used to calibrate DOLCE were extratropical sites.

DOLCE has also shown a weak performance at US-Twt, which is an irrigated rice paddy. This site gets flooded in spring and drains in early fall, then the rice is harvested. Only nine months were available for this site, which coincide with the flood and drain period between spring and fall. DOLCE could not depict the flooding and draining event, probably because none of the weighting products can represent such phenomena, so it is expected that the effects of seasonal flooding are not represented in DOLCE.

In this study, we sought a single weight for each product to apply globally. But we have reason to believe that different products are likely to perform better in different environments, so that different weights in different climatic circumstances might well improve the result of weighting overall. A similar suggestion was made in the studies of Ershadi et al. (2014) and Michel et al. (2016) who highlighted the need to develop a composite model, where individual models are assigned weights based on their performance across particular biome types and climate zones. We therefore tried to cluster flux tower sites into groups (such as vegetation type) so that each group maintains enough members to allow the in- and out-of-sample testing approach used above. We tried clustering by vegetation type, climate zone and aridity index, and implemented the same one site out-of-sample testing approach as above, but this time, in each cluster different sets of weights will be assigned to the weighting products. The motivation for this approach was to try to reduce the number of sites that did not show improvement as a result of the weighting, and ideally improve performance of DOLCE overall, yet none of these clustering approaches delivered any improvement, possibly due to the considerably reduced sample size in each cluster (increasing the likelihood of overfitting). It might also be a result of grid cell heterogeneity within the small sample that constitutes each vegetation type. Finally, as noted above, we have no guarantee that vegetation type is necessarily a good proxy for surface flux behaviour. A summary of this investigation and a plot showing the results of the clustered weighting by vegetation type (Fig. S2) are included in the supplementary material.

There are relatively few towers located in the Southern Hemisphere and the tropics (14 out of 159 sites) and none located in the dry climates over South West Asia and North Africa. The weighting was therefore mostly driven by the ability of products to match sites located in the temperate and cold zones of the Northern Hemisphere, so that performance in climate zones with low FLUXNET site density was under-represented when deriving DOLCE. This might raise questions about the performance of DOLCE in the tropics and the Southern Hemisphere. To evaluate DOLCE in these areas we calculated the four site metrics (RMSE, Mean bias, SD difference and Correlation) separately for two groups of sites, 1) those located in the Northern Hemisphere excluding Tropics and 2) sites located in the Tropics and/or Southern Hemisphere. We excluded the two sites ID-Pag and AU-Fog in this exercise since both are wetland sites, and so would complicate a determination of whether these two groups had notable behavioural differences. If systematic behavioural differences did exist between these two groups, we would expect relatively poorer performance of DOLCE at group2 sites compared to group1 sites. The results, shown in Fig. 10, appear inconclusive. DOLCE performed marginally worse at group2 sites overall, however with the limited number of

sites in group2, the validation of the performance of DOLCE in the tropics and Southern Hemisphere remains somewhat uncertain. The uneven distribution of eddy-covariance sites between the Northern and Southern Hemisphere and across the different climates might also explain why much of the largest seasonal differences DOLCE-MPI and DOLCE-LANDFLUX-EVAL shown in Fig. 6 and Fig. 7 reside in the low latitudes (tropics) and the Southern Hemisphere and the persistent

differences between DOLCE and LANDFLUX-EVAL in the tropics throughout the year. The expansion of the FLUXNET network into these areas that are lacking observations is clearly something that would improve DOLCE and LSM evaluation more broadly.

In this study, DOLCE was derived by combining remote sensing products that use an empirical approach (i.e. MPIBGC) and a more physical approach (i.e. MOD16, GLEAM v2A, GLEAM v2B and GLEAM v3A and PML). Constructing an ET product

by combining different approaches can take advantage of their desirable features and reduce their limitations (Zhang et al., 2016). This is indeed reflected by the enhanced performance of DOLCE over all the biome types against the reference products. Table 3 shows that the weights were attributed almost equally to each approach (i.e. ≈ 0.5 for both MPIBGC and the physical ensemble in tier 1 and tier 2), which indicates that the two approaches contributed equally in deriving DOLCE, it is not surprising that MPIBGC was the most weighted product since it is highly calibrated with flux tower data. In a further analysis,

we left MPIBGC out and we performed the out of sample tests using the five remaining products. We wanted here to test how the weighting will perform without MPIBGC. The plots in Fig. 11 and 12 show the results of 25 % out-of-sample test and one site out-of-sample test respectively. Overall, without including MPIBGC, the weighting offers a smaller performance improvement than that offered when MPIBGC is a member of weighting ensemble (Fig. 3 and Fig. 4 (a-d)). The distribution of the weights when MPIBGC is absent from the weighting is 0.3 for both PML and GLEAM_v3A, 0.2 for GLEAM_v2B,

0.13 for MOD16 and 0.07 for GLEAM_v2A.

It is important to mention that the weighting approach offers the possibility of enhancing DOLCE in the future by incorporating any ET dataset that becomes available, for example, WECANN, (Alemohammad et al., 2016), and HOLAPS v1.0, (Loew et al., 2016).

The limitations of the weighting approach presented here arise from issues in the observational data and limitations in the

weighted datasets that are employed. FLUXNET data inevitably has some instrumentation-related problems (Göckede et al., 2008) that affect the data quality of the measured flux in some terrains and under some environmental conditions. Moreover, FLUXNET sites are not globally-representative of all terrestrial ecosystems, and not evenly distributed across the biome types, which might lead to biases when computing the weights of the products in favour of the product that outperforms over the most frequent cover types. Also, in some areas where tower density is relatively high, the information from different towers is

not necessarily independent. Finally, a common imperfection in all the weighted products due to for example anthropogenic water management, will of course lead to the same imperfection in the derived product.

The uncertainty estimate presented here is firmly grounded in the spread of existing gridded ET products, but is better than this spread alone, since this spread has been recalibrated so that the uncertainty of DOLCE where flux tower data exists is precisely the spread of the recalibrated products.

## 5. Conclusion

In this study, we presented a new global ET product with monthly temporal resolution for 2000–2009 at 0.5° spatial resolution and a calibrated estimate of its uncertainty. The approach used to weight existing gridded ET products accounted for both the performance differences and error covariance between the participating ET products. The DOLCE product performs better than any of its six constituent members overall, as well outperforming other well-known gridded ET products (CSIRO-global, LandFlux-EVAL-ALL, LandFlux-EVAL-DIAG and PT–JPL) across a range of metrics. It was shown that despite the scale mismatch between the flux tower and the grid cell, the ensemble of flux towers as a whole does provide information about the grid cells that contain them, since the improvements delivered by the weighting approach were evident in sites not used to derive the weights. While the representativeness of the point scale for the grid scale is enhanced by only considering sites that lie within homogeneous grid cells we suggest that an optimal definition of homogeneity for flux behaviour be the subject of future investigation. Nevertheless, DOLCE appears to outperform existing gridded ET products overall, and offers the opportunity for improvement as more flux tower data and new gridded ET products become available. Expanding DOLCE over longer time periods and incorporating more diagnostic ET datasets (such as PT-JPL, CSIRO-global, GLEAM-V3B, SEBS, WECANN and HOLAPS) will be carried out in future versions.

## 6. Competing interests

The authors declare that they have no conflict of interest.

## 7. Data availability

DOLCE v1.0 can be downloaded from geonetwork.nci.org.au and its DOI is http://dx.doi.org/10.4225/41/58980b55b0495

## 8. Acknowledgements

Sanaa Hobeichi acknowledges the support of the Australian Research Council Centre of Excellence for Climate System Science (CE110001028). Gab Abramowitz and Jason Evans acknowledge the support of the Australian Research Council Centre of Excellence for Climate Extremes (CE170100023). This work used eddy covariance data acquired and shared by the FLUXNET community, including these networks: AmeriFlux, AfriFlux, AsiaFlux, CarboAfrica, CarboEuropeIP, CarboItaly, CarboMont, ChinaFlux, Fluxnet-Canada, GreenGrass, ICOS, KoFlux, LBA, NECC, OzFlux-TERN, TCOS-Siberia, and USCCC. The ERA-Interim reanalysis data are provided by ECMWF and processed by LSCE. The FLUXNET eddy covariance data processing and harmonization was carried out by the European Fluxes Database Cluster, AmeriFlux Management Project, and Fluxdata project of FLUXNET, with the support of CDIAC and ICOS Ecosystem Thematic Center, and the OzFlux, ChinaFlux and AsiaFlux offices. This work also used eddy covariance data acquired by the FLUXNET community and in

particular by the following networks: AmeriFlux (U.S. Department of Energy, Biological and Environmental Research, Terrestrial Carbon Program (DE - FG02 - 04ER63917 and DE-FG02 - 04ER63911)), AfriFlux, AsiaFlux, CarboAfrica, CarboEuropeIP, CarboItaly, CarboMont, ChinaFlux, Fluxnet - Canada (supported by CFCAS, NSERC, BIOCAP, Environment Canada, and NRCan), GreenGrass, KoFlux, LBA, NECC, OzFlux, TCOS - Siberia, USCCC. We acknowledge

the financial support to the eddy covariance data harmonization provided by CarboEuropeIP, FAO - GTOS - TCO, iLEAPS, Max Planck Institute for Biogeochemistry, National Science Foundation, University of Tuscia, Université Laval and Environment Canada and US Department of Energy and the database development and technical support from Berkeley Water Center, Lawrence Berkeley National Laboratory, Microsoft Research eScience, Oak Ridge National Laboratory, University of California -  Berkeley, University of Virginia.

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

**Tables**

**Table 1: Gridded ET products used in this paper.**

| ET product and Reference | Abbreviation | Time period & Spatial Resolution | Forcing data source | Calculation Method(s) |
|---|---|---|---|---|
| CSIRO-global (Zhang et al., 2010a) | CS | 1983–2006 0.5° Also available at 8km and 1° | Meteorological observations from flux tower distributed across all global biome types Remote sensing inputs | An extended ET product of CSIRO (Zhang et al., 2010b) that covers a global domain NDVI-based PM model PT equation for open water evaporation |
| GLEAM-V2A (Miralles et al., 2011) | G2A | 1980–2011 0.25° | Remote sensing based observations Gauged based precipitation | PT equation Canopy Interception Model, Soil water module and Stress module |
| GLEAM-V2B (Miralles et al., 2011) | G2B | 2000–2011 0.25° | Remote sensing based observations | PT equation Canopy Interception Module, Soil water module and Stress module |
| GLEAM-V3A (Martens et al., 2016) | G3A | 1980–2014 0.25° | Satellite based inputs Multi-source precipitation | A revised version of GLEAM V2A in which new satellite-observed geophysical variables have been incorporated and the representation of the surface soil moisture and evaporation has been improved |
| LandFlux-Eval-Diag | LFD | 1989–2005 1° | Simple mean of 5 diagnostic ET datasets | |

| | | | | |
|---|---|---|---|---|
| (Mueller et al., 2011, 2013) | | | | |
| LandFlux-Eval-All (Mueller et al., 2011, 2013) | LFA | | Simple mean of 14 Diagnostic, LSM and Reanalysis datasets. | |
| MOD16 MODIS global ET products (Mu et al., 2011) | MOD | 2000–2014 0.5° also available at 0.0 5° | Global Modeling and Assimilation Office (GMAO) meteorological reanalysis data Remote sensing inputs from MODIS 8-day retrievals | PM formula (Monteith J. L., 1965) |
| MPIBGC (Jung et al., 2011) | MPI | 1982–2011 0.5° | FLUXNET data from 253 sites Remote sensing datasets from (SeaWiFS) | Empirical methods: a Model Tree Ensemble (MTE) Machine learning techniques |
| PML PM-Leuning model (Zhang et al., 2015) | PML | 1981–2012 0.5° | GMAO Reanalysis products | PM Leuning method |
| PT–JPL (Fisher et al., 2008) | PT | 1984–2006 1° | Meteorological reanalysis data from ISLSCP –II Remote sensing based observations from monthly AVHRR data | PT equation |

**Table 2: Four metrics (RMSE, Mean bias, SD difference and Correlation) of DOLCE at three irrigated sites, and the number of available monthly records for each site.**

| Site-Code | Longitude | Latitude | Description | RMSE | Mean bias | SD difference | Correlation | Number of months |
|---|---|---|---|---|---|---|---|---|
| US-Ne1 | -96.4766 | 41.1651 | Rice paddy | 16.6 | -7.41 | -9.25 | 0.96 | 103 |
| US-Ne2 | -96.4701 | 41.1649 | Mead irrigated continuous maize site | 15.8 | -5.05 | -7.44 | 0.95 | 103 |
| US-Twt | -121.6521 | 38.1055 | Mead irrigated maize-soybean rotation site | 91.9 | -67.39 | -55.23 | 0.49 | 9 |

**Table 3: (1) Bias of weighting products and (2) weights assigned to the bias corrected products in the case of each of the three DOLCE tiers, and the number of flux tower sites used to feed the weighting.**

| Product | DOLCE tier 1 138 sites | | DOLCE tier 2 151 sites | | DOLCE tier 3 159 sites | |
|---|---|---|---|---|---|---|
| | weight | Bias $W\ m^{-2}$ | weight | Bias $W\ m^{-2}$ | weight | Bias $W\ m^{-2}$ |
| MOD16 | 0.041 | 3.756 | 0.05 | 4.0 | | |
| MPIBGC | 0.495 | 3.837 | 0.537 | 3.882 | | |
| GLEAM –v2A | –0.026 | 6.180 | 0.123 | 6.098 | 0.44 | 5.735 |
| GLEAM-v2B | 0.192 | –3.571 | | | | |
| GLEAM-v3A | 0.171 | –5.483 | 0.151 | –5.221 | 0.56 | –6.344 |
| PML | 0.127 | 4.982 | 0.139 | 4.668 | | |

**Figures**

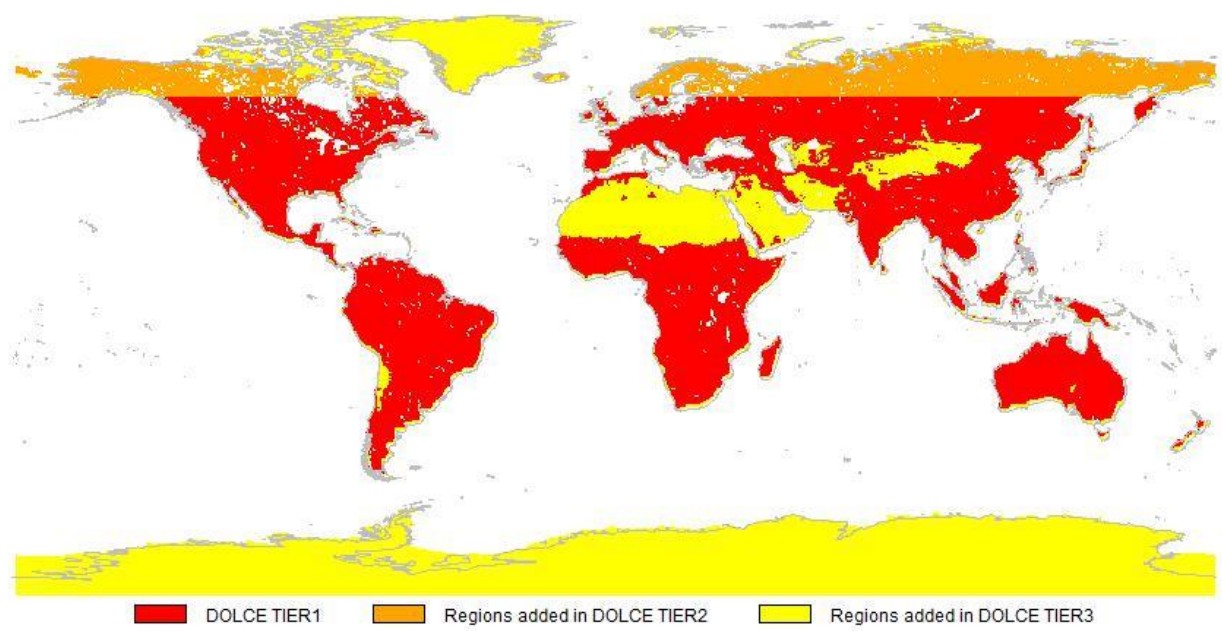

Figure 1: Spatial coverage of DOLCE tier 1 (red), regions added in tier 2 (orange) and tier 3 (yellow). Spatial coverage varies temporally; May 2000 is shown here as a representative example.

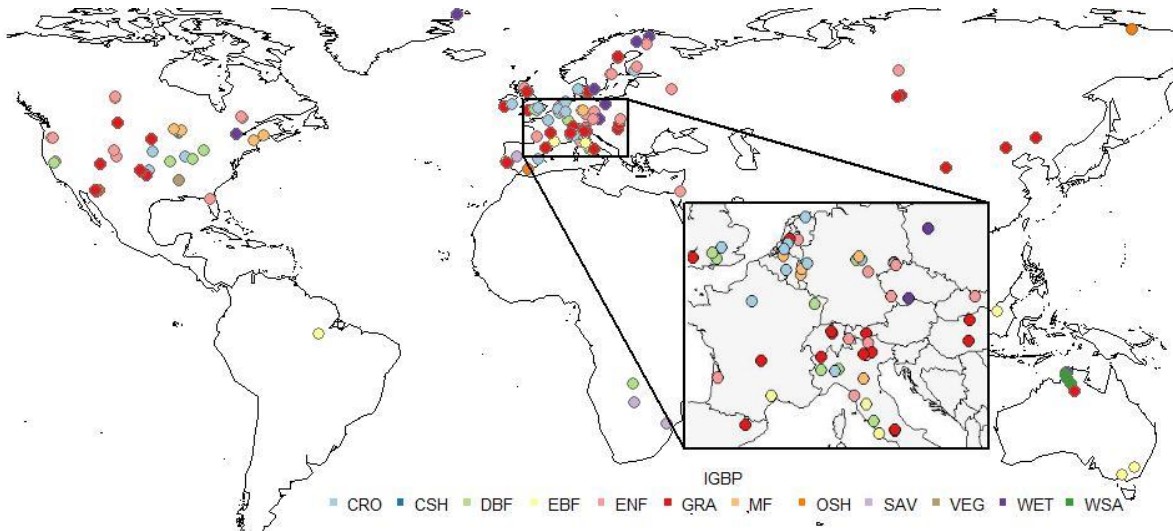

Figure 2: Location of the 159 flux tower sites involved in building DOLCE, classified by vegetation type given by International Geosphere-Biosphere Programme (IGBP): Croplands (CRO), Closed Shrublands (CSH), Deciduous Broadleaf Forest (DBF), Evergreen Broadleaf Forest (EBF), Evergreen Needleleaf Forest (ENF), Grasslands (GRA), Mixed Forests (MF), Open Shrublands (OSH), Savannas (SAV), Cropland/Natural Vegetation mosaic (VEG), Wetlands (WET), and Woody Savannas (WSA).

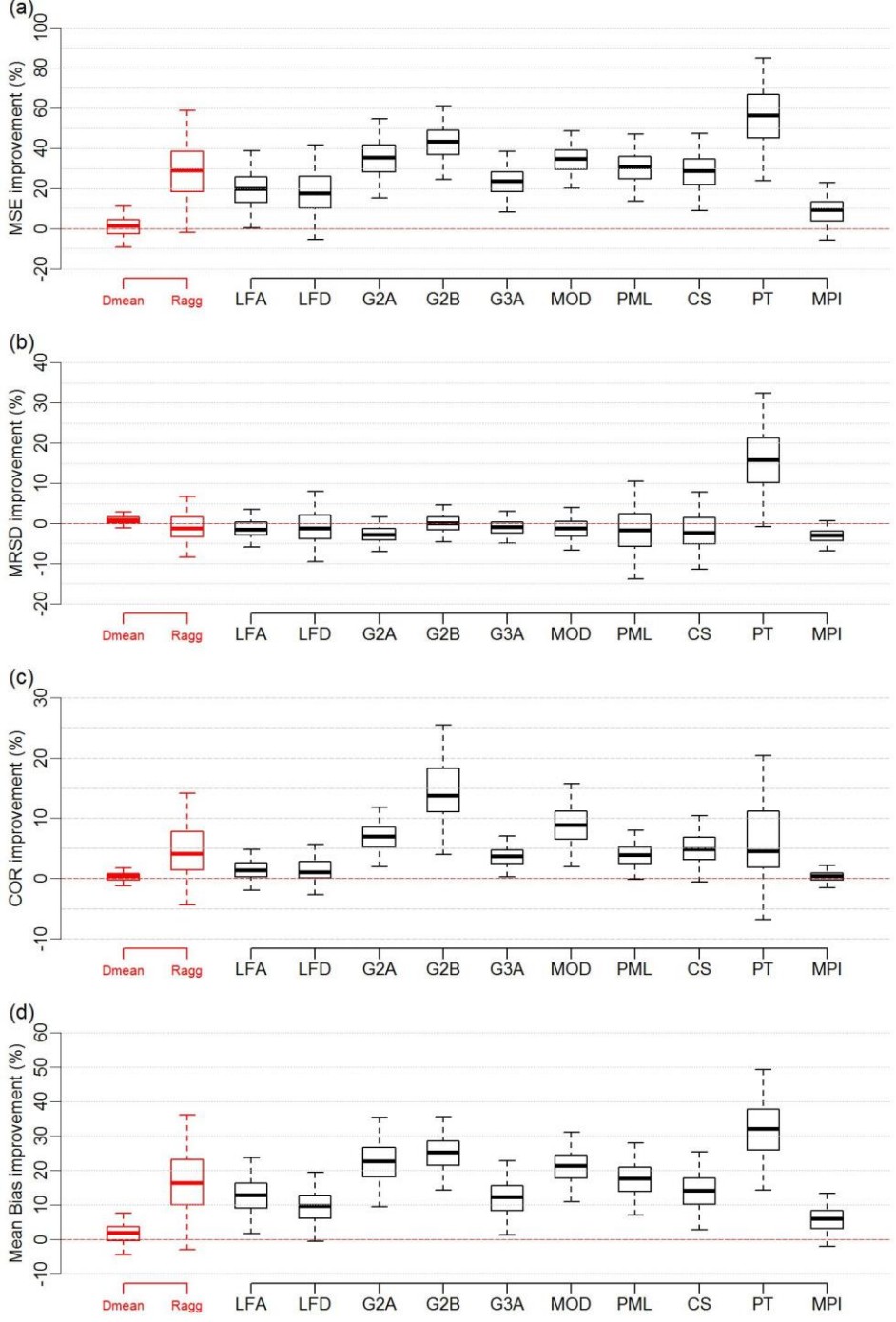

**Figure 3: Box and whisker plots displaying the percentage improvement that the weighted product provides in the 25% out-of-sample sites test for four metrics: MSE (a), MRSD (b), COR (c) and Mean bias (d), when compared to the equally weighted mean of the Diagnostic Ensemble (Dmean), aggregated Reference Ensemble (Ragg) and each member of the reference ensemble. Box and whisker plots represent 5000 entries, each entry is generated through randomly selecting 25% of sites to be out sample.**

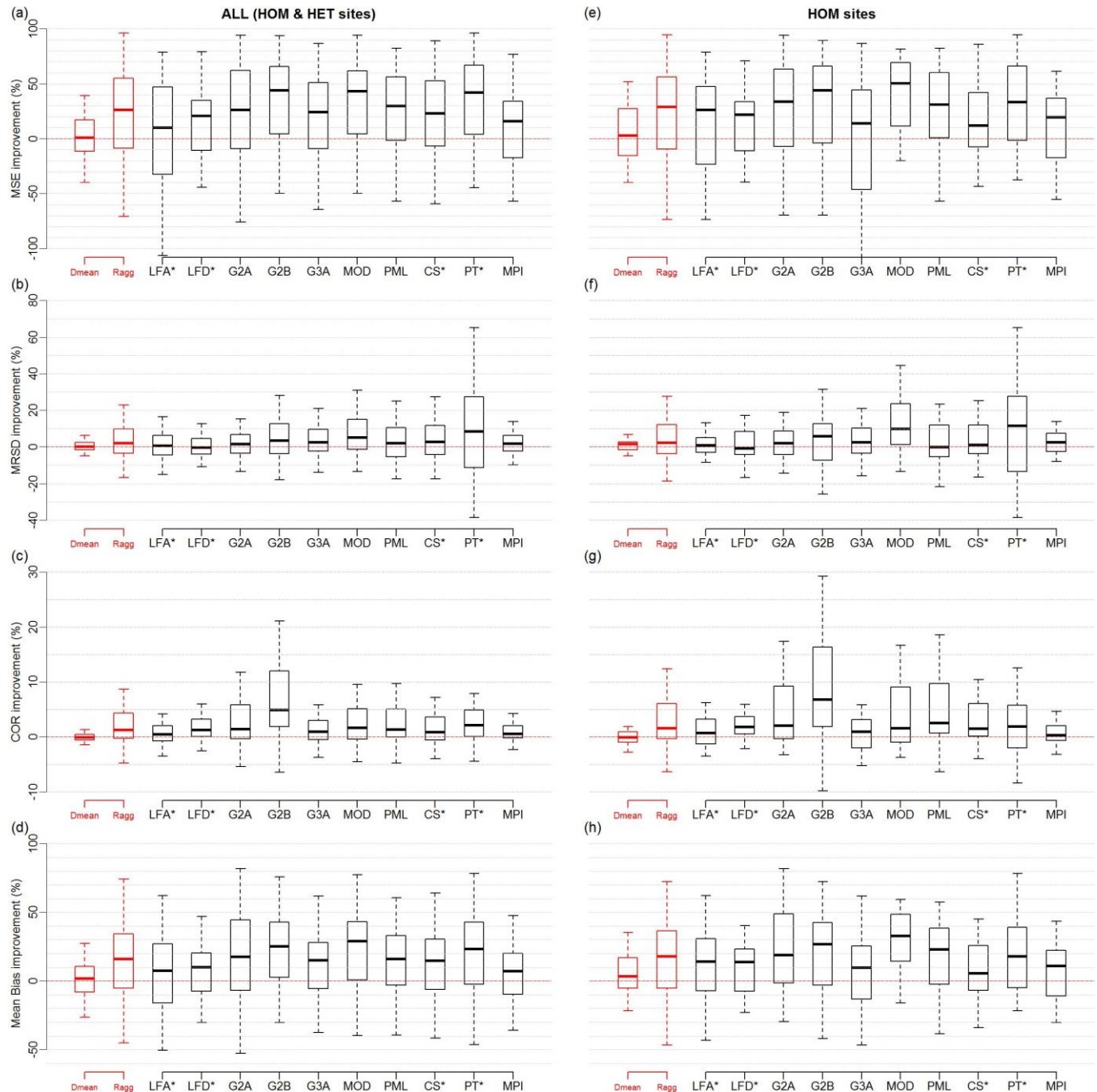

**Figure 4: In (a), (b), (c) and (d), as for Fig. 3 but showing the one site out-of-sample test. Box and whisker plots are generated through selecting one site to be out sample and are repeated for all 138 sites. Products marked with * have limited spatiotemporal availability relative to the diagnostic ensemble, and testing against the LFA, LFD, CS and PT products was limited to 110, 108, 108 and 72 sites respectively. In (e), (f), (g) and (h), the one site out-of-sample test is trained by HOM-case sites data only.**

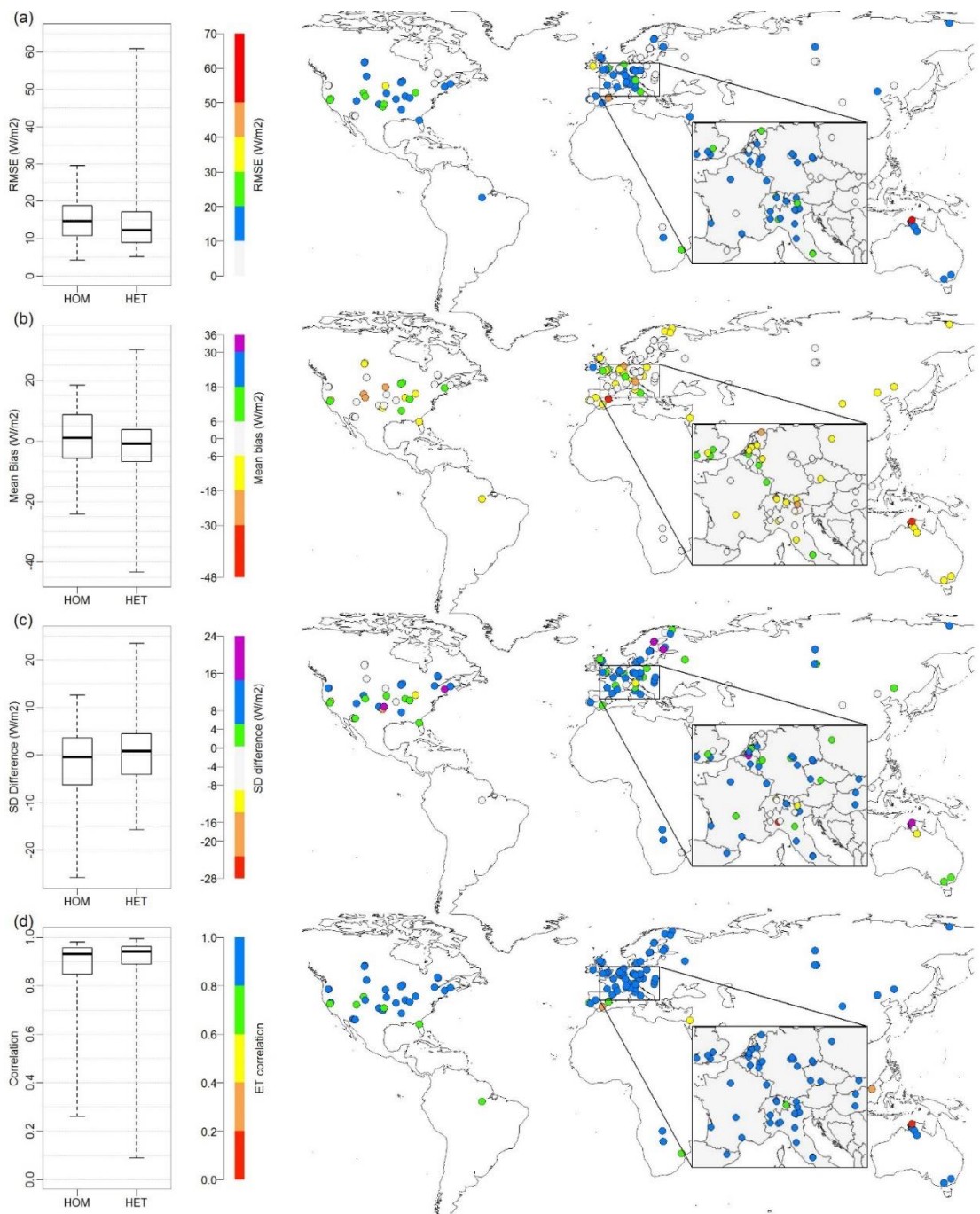

**Figure 5: The global maps in the right column show the performance of DOLCE against in-situ measurements at each of the 142 sites by calculating four statistics: (a) RMSE, (b) Mean bias, (c) SD difference and (d) Correlation. The boxplots on the left display the results of these four metrics calculated separately for the sites that are satisfying the HOM-case and the HET-case.**

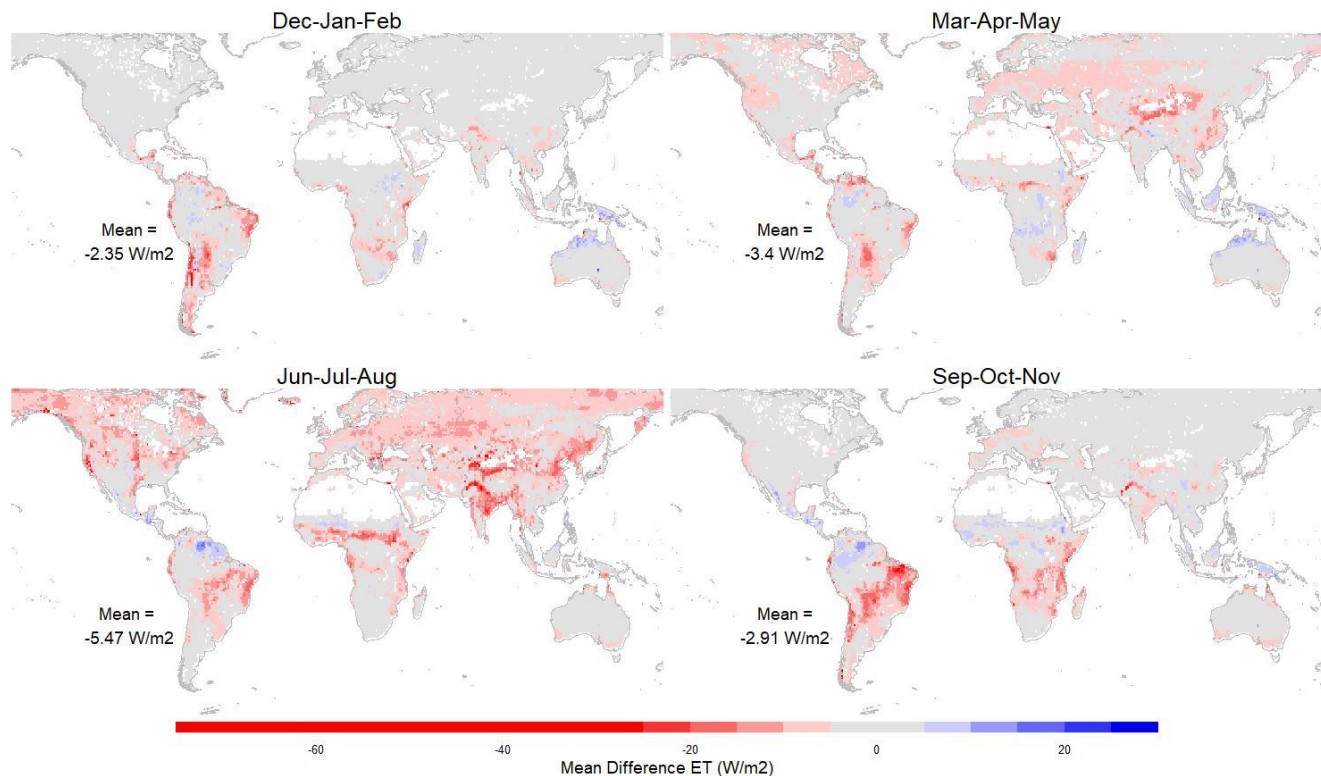

**Figure 6: Difference between the seasonal global ET from MPIBGC and DOLCE calculated over the period 2000–2009.**

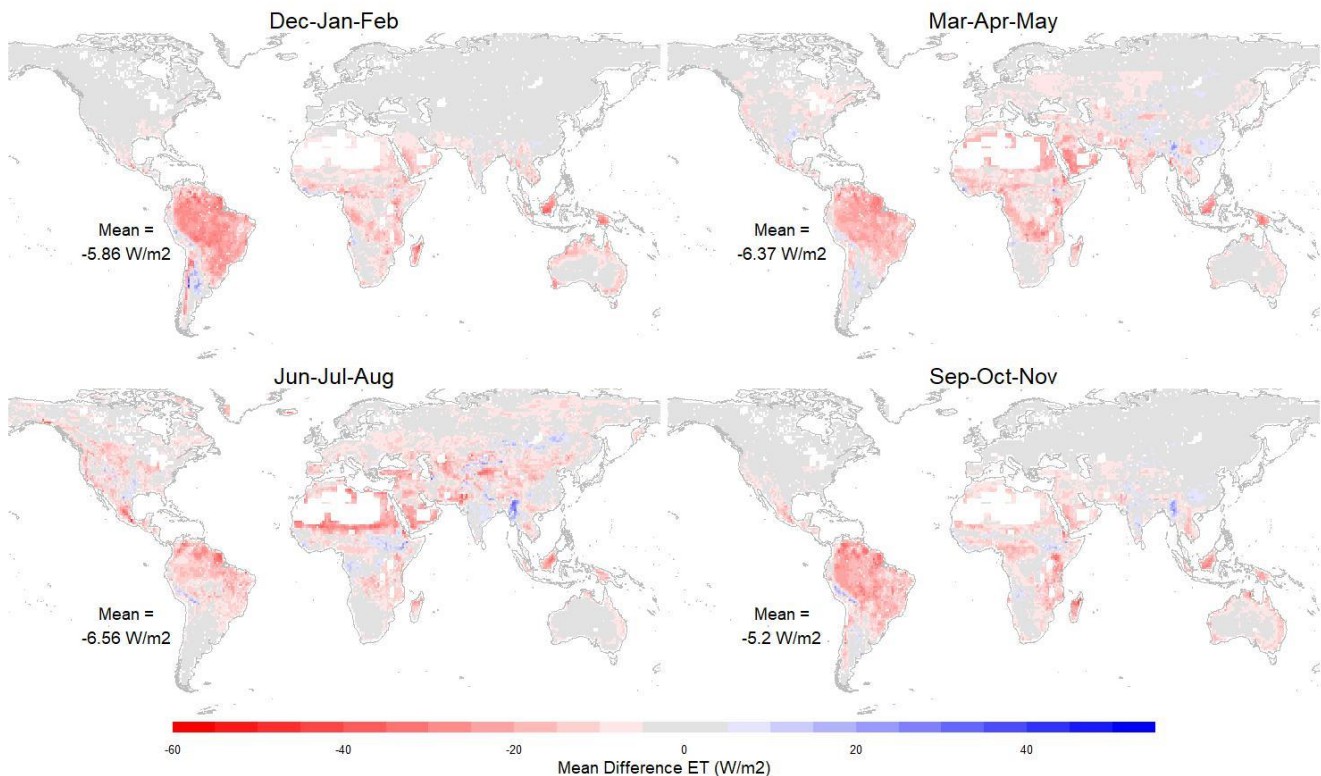

**Figure 7: Difference between the seasonal average ET from LandFlux-EVAL-DIAG and DOLCE calculated over the period 2000–2005.**

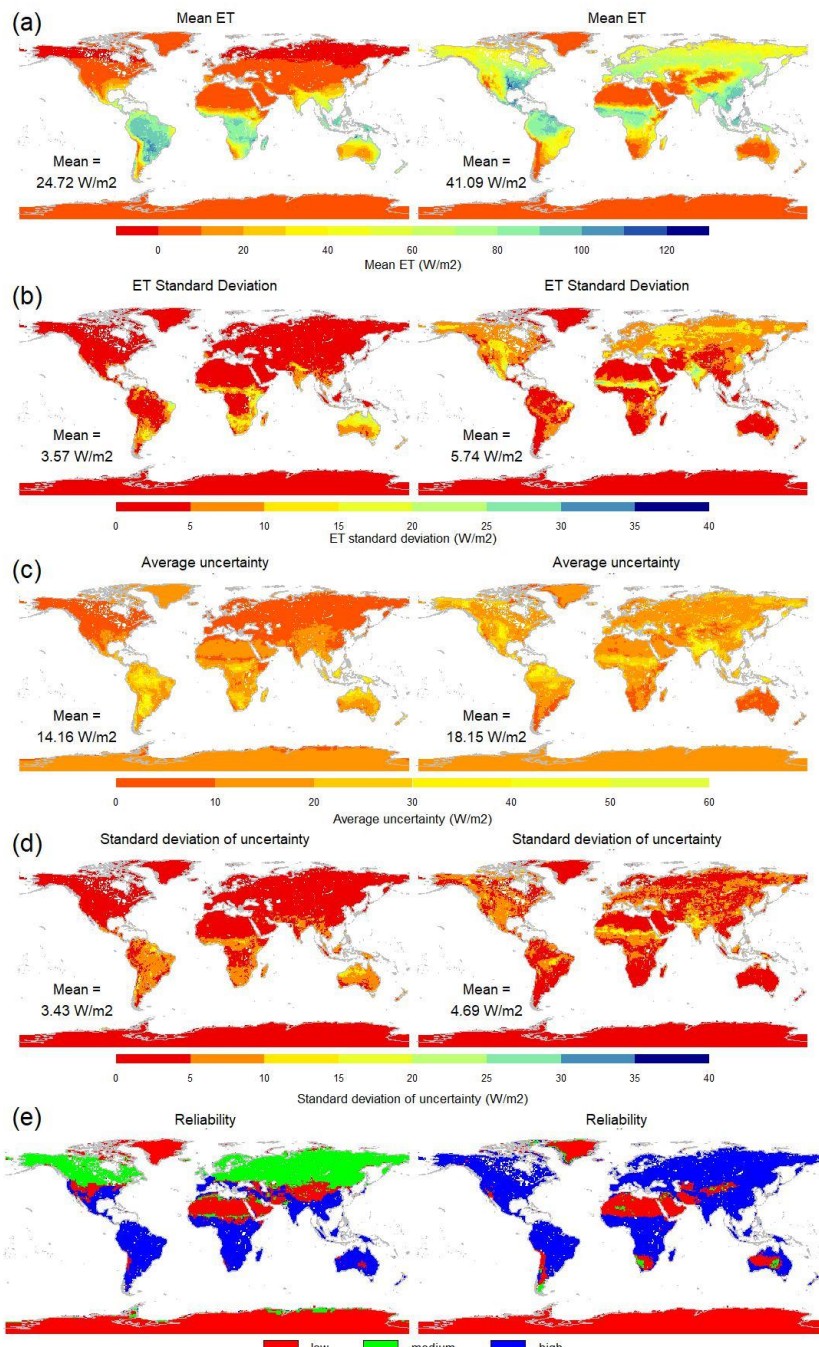

**Figure 8: Seasonal (a) global ET and (b) its variability (standard deviation), (c) time average of uncertainty (the standard deviation uncertainty $\sqrt{\sigma_e^2}$ shown in Equation 11) (d) standard deviation of uncertainty over time (e) reliability, defined as high ($\frac{Uncertainty\ SD}{mean\ ET} \leq 1$ in blue), medium($|mean\ ET| \leq 5, Uncertainty\ SD < 10$ and $\frac{Uncertainty\ SD}{mean\ ET} \geq 1$ in green) and low (in red). DJF is shown in the left column and JJA in the right column. The global mean values in (a-d) are area weighted.**

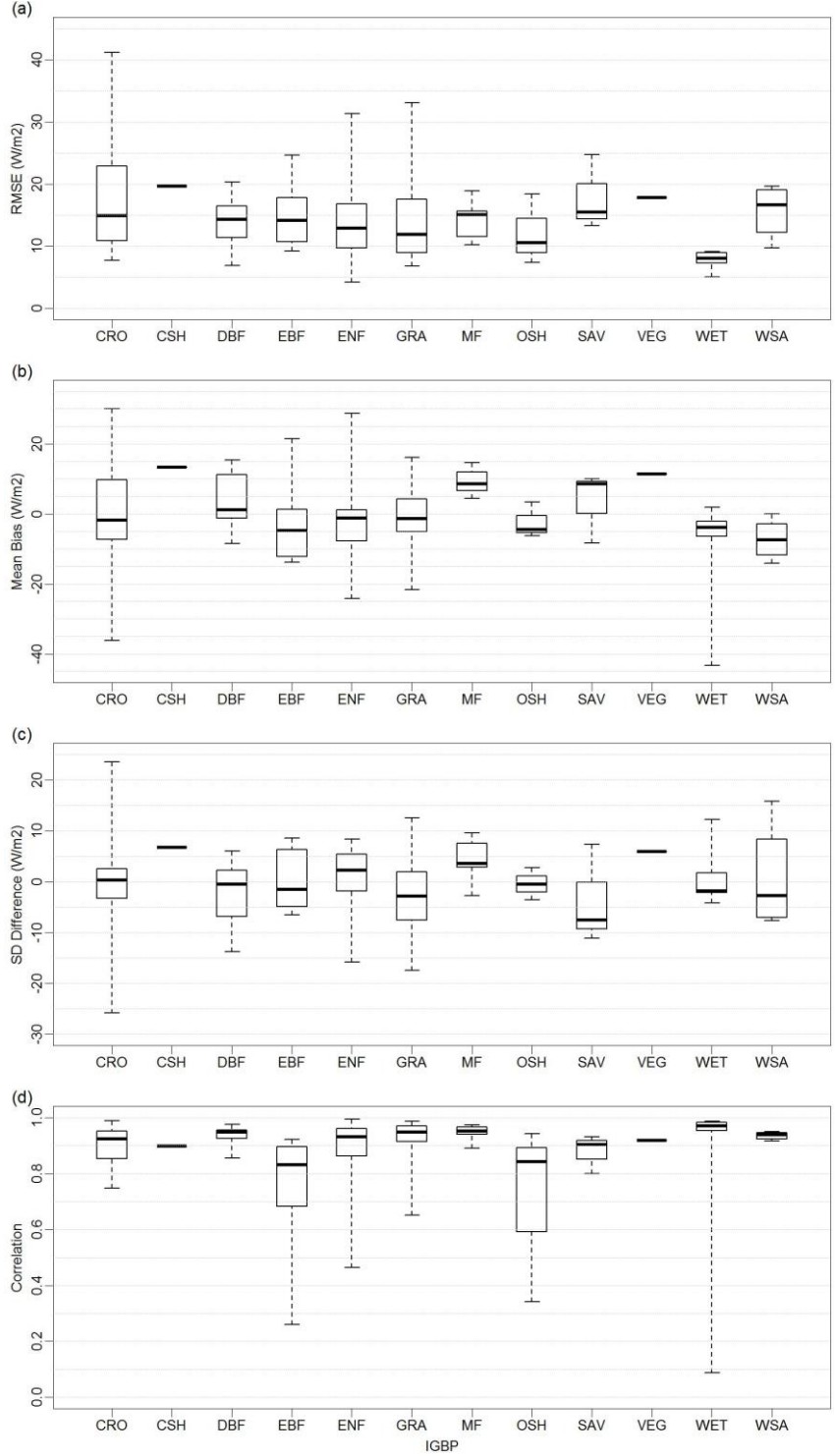

**Figure 9: Four statistics, (a) RMSE, (b) Mean bias, (c) SD difference and (d) Correlation, calculated for DOLCE at 142 flux tower sites and displayed by biome types. See Fig. 2 for biome abbreviations.**

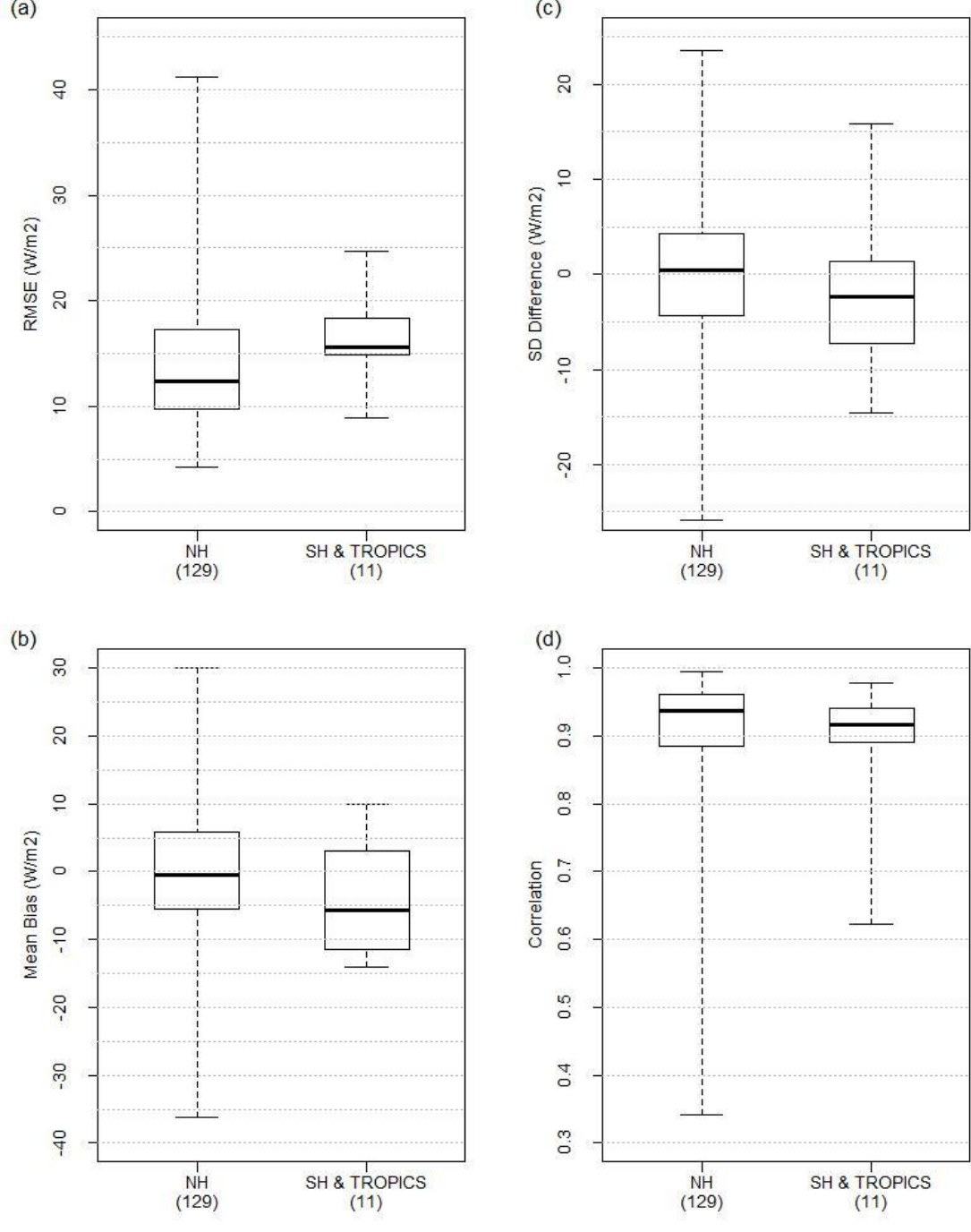

**Figure 10: Four statistics, (a) RMSE, (b) Mean bias, (c) SD difference and (d) Correlation, calculated for DOLCE separately at 129 flux towers located at the Northern Hemisphere excluding the tropics (NH) and at 11 towers in the Southern Hemisphere and the tropics (SH & TROPICS).**

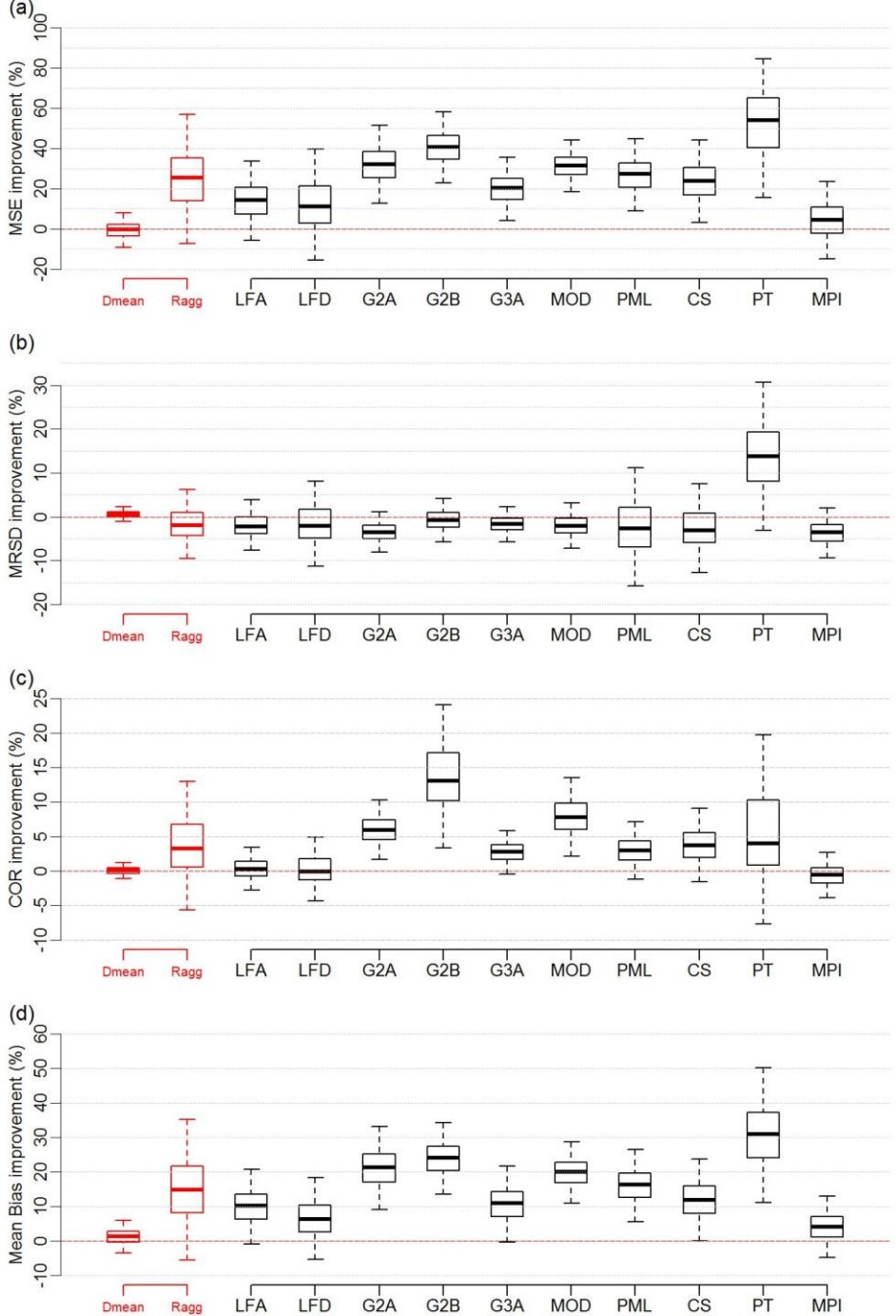

**Figure 11: Box and whisker plots displaying the percentage improvement that the weighted product excluding MPIBGC provides in the 25% out-of-sample sites test for four metrics: MSE (a), MRSD (b), COR (c) and Mean Bias (d), when compared to the equally weighted mean (Dmean) of the Diagnostic Ensemble, aggregated Reference Ensemble (Ragg) and each member of the reference ensemble. Box and whisker plots represent 5000 entries, each entry is generated through randomly selecting 25% of sites to be out sample.**

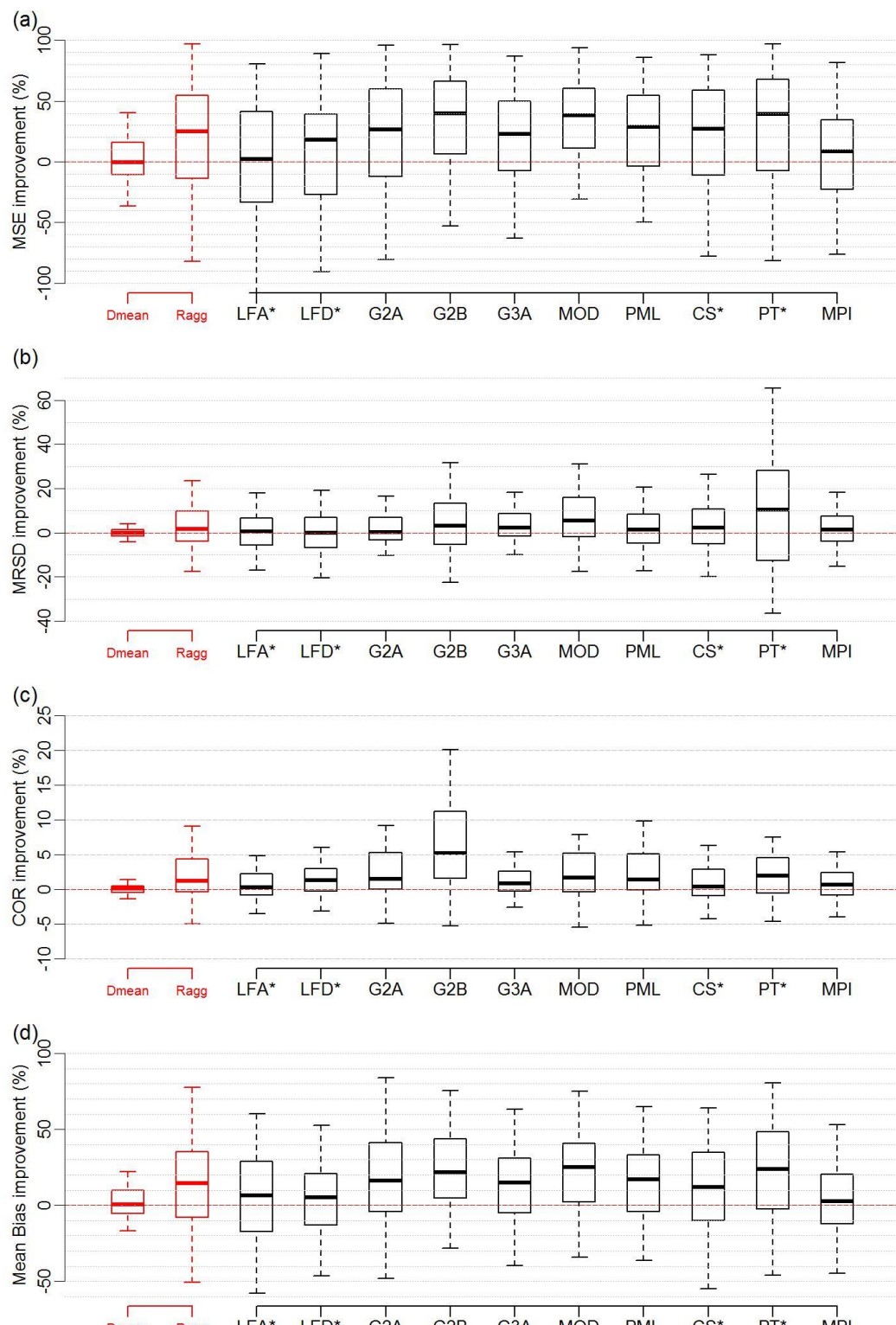

**Figure 12: Box and whisker plots displaying the percentage improvement that the weighted product excluding MPIBGC provides in the one site out-of-sample test for four metrics: MSE (a), MRSD (b), COR (c) and Mean Bias (d), when compared to the equally weighted mean (Dmean) of the Diagnostic Ensemble, aggregated Reference Ensemble (Ragg) and each member of the reference ensemble. Products marked with * have limited spatiotemporal availability relative to the diagnostic ensemble, and testing against the LFA, LFD, CS and PT products was limited to 110, 108, 108 and 72 sites respectively.**