# Peer review of "Derived Optimal Linear Combination Evapotranspiration (DOLCE): a global gridded synthesis ET estimate"

_Hydrology and Earth System Sciences, 2017_

## Referee Comment (RC1) · P. Dirmeyer (Referee) · 3 May 2017

The authors present a method to more optimally combine global ET estimates using FLUXNET sites as anchor points for cal/val (which they name DOLCE), producing a demonstrably better product when validated at FLUXNET sites (even with cross-validation), although there are many instances where DOLCE is outperformed by one or more other products at individual sites. The approach presented here is limited to providing a single weighting function per input data set (see Table 3) and is limited to a 10-year period. The approach is flexible and can incorporate other / more input data sets and validation sites easily.

This is an interesting, useful and worthy paper, demonstrating real improvements but

also casting light on the difficulty of producing vastly improved globally distributed estimates of ET. I very much appreciate the attempts to identify the effects of misrepresentation of vegetation classes. I do have a number of comments:

General comments:

1. There is obviously a question regarding the representativeness of the FLUXNET stations to their regions (grid box average or dominant surface conditions), which the paper addresses fairly well. However, one specific issue jumps out at me: what about rare representatives, particularly in the tropics? FLUXNET is woefully thin on stations at low latitudes. The out-of-sample tests are like an OSSE, but there is little sampling in the tropics to play with here. What was the specific effect of denying low-latitude stations? Can we use this study to make case to prioritize FLUXNET expansion into the tropics, S/SW Asia and the Southern Hemisphere? I would like to see this issue discussed more, where the global maps are presented and also in the Discussion. Motivating targeted FLUXNET network expansion would be a good "broader impact" of this study.

2. What about the uncertainty/error in the FLUXNET observations? They certainly contain random errors and systematic biases. For a measured quantity that has a red-noise spectrum (following a Markov process) the random error can be estimated from the behavior of lagged autocorrelations (cf. http://dx.doi.org/10.1175/JHM-D-15-0196.1). Random error will systematically degrade correlation and affect other statistics (cf. http://dx.doi.org/10.1175/JHM-D-15-0063.1). Systematic error is more difficult to identify, but the energy balance corrections in FLUXNET2015 give some clue (see specific comments below). All these mean that using FLUXNET for cal/val is itself flawed and imperfect. On the other hand, model ET is inherently precise (no random error) and can be used to estimate/differentiate this type of error from others by noting its statistical differences from the instrument records. This issue should be acknowledged and discussed, including how the assumptions in DOLCE regarding errors (that they are uncorrelated) affect results. In other words, more discussion about uncertainties.

3. In the Supplement: There are captions for Fig S1 and Fig S2, but the PDF does not contain any figures!! Please recheck the rendering.

Specific comments:

P3 L6: Change "85 FLUXNET tower data" to "85 FLUXNET towers"

P3 L8: "ground-truthed" is not an appropriate verbification for this context. Change '...gridded ET data sets are "ground-truthed" using flux tower data from FLUXNET...' to '...tower data from FLUXNET provide ground truth for gridded ET data sets...'

P5 L16: Change "where" to "were"

P7 L22: RSD cannot convey whether the variance is too large or too small. So for the changes shown in Fig 3b - we cannot tell whether the improvement is due to an increase or decrease in standard deviation. Furthermore, what about locations where mean ET (denominator) is near zero - does that explain some of the very large values and changes?

Fig 3: So the spread is across 5000, and each of those is an average of 25% out-of-sample stations, right? It is not 5000x172x0.25 points. Please make clear. Also, Fig 4 suggests individual o-o-s stations frequently fare worse. Transferability of calibrations appears to be kind of weak, which is not a surprise. Calibration transferability is a very difficult enterprise. This should also be acknowledged, either here or in the Discussion section (wishing PILPS-San Pedro had been completed as it would have really shone a light on this problem).

P9 and Fig 4: "widespread out-of-sample improvement that this approach offers over existing gridded ET products" seems like a bit of an overstatement - there are frequently locations that fare worse, and sometimes much worse. In Fig 4, there is typically a tremendous range, and usually the central two quartiles encompass the zero line. For RMSE it appears that ~20-49% of the time the estimates are worse than other

products and methods, 30-55% for RSD, and 20-55% for COR. While there is definitely a net (and welcome) improvement in almost all cases, often the DOLCE estimate is worse. This should be acknowledged.

P9 L26: "Standard Deviation (SD) difference" - this is clearly not the same as RSD defined earlier - please define, which minus which?

Table 2 has little accompanying discussion, and what is there is very shallow adding little to comprehension. Please discuss more or remove the Table if it does not warrant discussion.

Fig 6: Here it could be said you use DOLCE to estimate MPI, and most places have a positive bias. But the energy-balance-corrected fluxnet data, which close the surface energy balance by construct conserving Bowen ratio, consistently increases ET compared to the raw measurements (at 107 of 122 FLUXNET2015 Tier-1 stations during JJA by my quick calculation). There is recent independent indication that tower sites bias low because of errors in the turbulence theory applied to estimate fluxes (see: http://dx.doi.org/10.1002/2017GL073499). Could FLUXNET instrument error (and the simplicity of the energy closure correction) contribute to systematic biases in DOLCE? Please discuss here or in the Discussion section.

Re Fig 7: Much of the largest differences are at low latitudes where there is little FLUXNET data for calibration. Please discuss, as I see this as a major issue (more with FLUXNET station distribution than your methods, but the problem of representative ET estimates in the tropics is an ongoing concern).

P10 L21: "ET doesn't exhibit any seasonal change over Greenland and the deserts in North Africa..." - not expected in the absolute, because mean values are tiny. It would be more informative to also show relative (percentage) changes, which are more relevant for local water balances.

Discussion §: Please also speculate whether some spatial variability in weighting could

improve estimates further, even if only in 2 or 3 categories of weights.

Fig 9: Only stations in the tropics are 2 EBF stations; the savannah stations are in southern Africa. These are not o-o-s results, right? I would have expected these types to stand out more, but perhaps the samples are biased to the extratropical stations in the categories. Thoughts?

Also Fig 9: Crops are tricky. The category is a catchall that is unsatisfactory because there is such variability in phenology, seasonality, stomatal resistances, etc. Are any of the CRO stations rice (which acts very different because of seasonal flooding). Not surprising many of the biggest errors are there.

P13 L1: "...for example anthropogenic water management..." - but it was stated earlier that irrigated sites were excluded. Please expand on this comment.

Conclusion: Please also state plans for updates, future versions, perhaps (hopefully) covering a longer time period!?

References: There seem to be a lot of redundancies in author lists for papers with names showing up 2 or 3 times for the same entry. Please check.

---

## Short Comment (SC1) · 6 May 2017

I read with great interest this study about merging different terrestrial evaporation (ET) products. Given the current uncertainty of existing ET estimates, and that it is not yet demonstrated that a single methodology outperforms the others, avenues to merge products need to be explored, and I'm very glad to see efforts in that direction.

During my reading I took some notes, which I would like to share with the authors with the hope that they could contribute to the paper revision, or to future developments of DOLCE.

Product selection

While acknowledging the difficulties of finding the rights products to derive a synthesis product, I am a bit surprised about some of the choices. As the authors state, product selection should follow the criteria of product diversity, so ideally single algorithms with different strengths and weaknesses should be combined together. In that sense, I would have not considered an already synthesis product such as LandFlux-Eval as a possible candidate for the merge. In my opinion, a more valid alternative regarding LandFlux efforts could have been the three single products publicly available based on different ET algorithms (https://hydrology.kaust.edu.sa/Pages/GEWEX_Landflux.aspx). Also, I do not see much interest in combining obsolete versions of products with the current, and presumably better, product, as it has been done for GLEAM. Just GLEAM V3A (and perhaps GLEAM V3B) seems to me a better option.

An interesting product is MPI. This is a global extrapolation of the tower fluxes, the same tower fluxes that are used to decide on the merging weights, and quite different to the other products, which we could consider more "physically" based and less "calibrated" with the tower data. It is not surprise then that MPI is by a large margin the more weighted product. I do not deny that it can be a valid product for the merge, although much less independent than the other products with respect to the tower data. In that sense, it could have been very informative also to see how the merging works, and how the weights are distributed, when that product is left out.

Spatial and temporal resolution

The resolutions given in the Table 1 seem wrong for a number of products (e.g. MODIS original resolution is 1 km, Zhang 2010 is 0.05 deg, MPI and Zhang 2015 0.5 deg). Also, the periods of available data should also be revised for GLEAM (I think GLEAM V2A covered 1980-2011 and V3A 1980-2014).

As far as I can see, the only datasets limiting the study period to 2000 is MOD16 and GLEAM V2B. Perhaps the products going into merging could have been separated

in two groups, a bit similar to what has been done regarding geographical coverage: from a much earlier year than 2000 will all but MODIS and GLEAM V2B, and from 2000 including MODIS and GLEAM V2B. That would have resulted in a much longer DOLCE dataset, presumably based on a larger collection of tower data and more ET products.

A shorter time period than monthly will result is a more useful product. Daily will a be a better objective for future developments, although it will require a different selection of products, possibly more based on the "physically" based diagnostic ET products, where daily is a common time scale. I would suspect a more complex merging exercise, given the larger amount of time variability that needs to be captured by the merge product. ET and FluxNet datasets

Tier 3 (i.e, Greenland and Antarctica) is just a very close weighting of an obsolete version of GLEAM and a newer version. It seems a bit awkward to distribute that as part of a synthesis product. It may have been better to just remove those regions from the synthesis product given that nearly no one dares to estimate ET over there (understandably).

Some of the ET datasets considered are based on algorithms that estimate separately interception, evaporation from the canopy, and evaporation from the soil (e.g, MOD16, PT-JPL, and GLEAM). Under the assumption that routine EC observations perform very poorly for rainy conditions, in principle interception is not captured by the tower observations. In some recent ET evaluations of these products the tower data has been filtered to remove rainy periods and the interception component has not been evaluated. A discussion about this could have been interesting, given that, as far as I can see, the tower data is not filtered for precipitation conditions, and the merged product is a total ET product.

Regarding the energy closure issue, the text may give the impression that fluxes correction is always possible, but a large number of stations do not measure Rn and/or G. Given that only corrected fluxes are used in the study, I imagine that a number of sta-

tions have to be discarded as they corrected fluxes were not available (FLUXNET-2015) or could not be estimated (LaThuille-2007). This may be worth mentioning.

Merging technique

If I understand the method correctly, the weights are global (i.e. one value for all pixels) , time-invariant (i.e., an annual value), and the bias-correction is what Bishop, 2013 calls a "global bias" correction (i.e., is a single value per product using all towers in the in-sample training dataset). If this is true, if Product A was performing better than product B over some biomes and/or at some periods, the method cannot be used to weight more or less the products to reflect that difference in performance. If I am correct, I wonder if there is a way to modify the weighting to take into account those differences. We typically see that ET products perform differently at different biomes and/or seasons, so it may be advantageous to capture this in the weighting. Bishop, 2013 was quite illustrative about this. Given that for that temperature example the "truth" was quasi-global (i.e., not over a very few pixels like for the ET), just a "per-cell bias" correction, even without weighting, outperformed the weighted product with a prior "global bias" correction. Of course, the "per-cell bias" correction and weights cannot be be applied here, given the limited geographical coverage of the flux "truth", so the problem is more complex.

I guess your first take at this is your sites clustering based on vegetation type, presented in some detail in the Supplement. You conclude that it did not improve overall the DOLCE performance based on a global analysis, but I would be interested to see the results (in the Supplement I can download the figures summarizing the analysis are missing). Perhaps other schemes that better cluster the flux behavior are worth investigating for future versions of DOLCE.

Results

It is stated that the MSE plot in Fig. 3a shows the MSE of the weighted product being better than the ensemble mean, but I do not see it. The central line of the first whisker
box in Fig 3a is at the zero line. Perhaps I am missing something regarding how to read these plots. As a side note, it may be good to say what the end of the whiskers represents. In most occasions it is used to represent the max and min of the data, but it is not always the case.

In the same Fig. 3a, the whisker boxes for the individual products make me think again about the MPI product. Based on the "heavy" calibration of MPI with the same tower data used to derive the weights, I would speculate that if the MPI product was removed from the merging, the percentage improvements of this new weighted product (i.e., without MPI) over the individual products will be much smaller. This may give a different perspective of the exercise regarding the skill of the tower-based merging to combine the more "tower-independent" physically based products.

If we just concentrate on the improvements of the weighted product with respect to the equally weighted product (i.e., first whisker box in Fig 3 a-b-c), the gain in performance of the weighted product seems small. Again, if I read these plots correctly, the gain for MSE and COR is minimal, only the RSD shows some improvement. But given the definition of the RSD metric, I wonder is this is mostly associated to the bias correction. If I understand this correctly, after the bias correction mean-dataset and mean-observation will be equal, so RSD is abs(sigma-dataset –sigma-observation). In other words, I am wondering about a comparison of the equally weighted product, but with a bias-correction first, and the weighted product. I am assuming here that the equally weighted mean did not involve a prior bias correction, as nothing was stated in the paper, but I may be wrong.

In the HOM and HET comparison, I see very small improvements in MSE, larger for RSD, and not much for COR (the median of the whisker box is for MSE and COR is at the zero percentage line). And I wonder if the separation into HOM and HET sites may have also implied a separation in land covers, so the improvements we see are more related to the weighted product working better for some specific biomes. One may think that land covers such as forested areas are more likely to be represented in

the HOM class, compared with e.g. croplands. I wonder if this has been checked, i.e., that the biome representation in HOM and HET classes does not change too much.

Regarding the boxplots of Figure 5, it is true that the end of the whiskers are larger for the HET sites, but the RMSE and correlation median and percentiles look slightly better for the HET class. I think this is also worth discussing, as it can potentially indicate again that the differences in performance between HOM and HET classes are small, with just a few HET sites having bad statistics. Given that this is based on one site out-of-sample, I wonder if the bad performance at some individual sites may be nothing to do with homogeneity, but with the fact that the site is one of a kind, so any weights derived from a sample without that site are not informative. Given the location of some the sites, it would not be a surprise.

MPIBGC is the larger contributor to DOLCE, with a weight close to ∼0.5. Perhaps it is not surprising than the differences of MPI with DOLCE sown in Figure 6 are smaller than with LandFlux-EVAL-Diag in Figure 7, which is not part of DOLCE.

I acknowledge that it is quite difficult to come out with the right classes to try to illustrate the reliability of the weighted product. But when I look at figure 8-c what I broadly see is that arid places, Greenland and Antarctica have low reliability, snowed places medium, and the rest high. Perhaps just the uncertainty over the mean ET would have been more informative.

Discussion

The discussion about a possible selection of HOM sites to construct DOLCE is quite appropriate. Being very strict, and considering that the typical tower fetch is of the order of hundreds of meters, while the grid cell has an area of ∼2500 km2, possibly none of the sites will truly qualify as HOM. But if we want to keep using tower fluxes, we need to live with this.

Perhaps a simple measure to reduce the effect of this tower-fetch and cell-scale mismatch is to try to work at finer resolutions in the future. GLEAM, MOD16 and Zhang, 2010 are already at resolutions 0.25 deg. PT-JPL will be soon available at 5 km. Perhaps a couple of products will never be available at 0.25 deg (e.g., MPI), but they could be downscaled to 0.25 deg and merged with the other products (e.g., by a nearest-neighbor interpolation if we do not want to add any extra information).

The lack of success of any clustering of the sites based on vegetation, climate, etc, possibly is more indicative of the limitations of the tower flux, rather than limitation of the conceptual idea. Even if not terribly successful, a look at how the relative weights of the different products change for different clusters can be informative regarding the products performance for different conditions.

As I mentioned before, using MPI as part of the weighted product is perfectly valid. But I think the main feature defining MPI for this study is not the fact that it is a statistical product, but more that it is a global extrapolation of the same tower fluxes used to derive the weights. As clearly stated in the paper, the tower fluxes have limitations, which will have an impact on the weighted product. Presumably, the MPI product will suffer from the same limitations than the tower fluxes, but these limitations will not become apparent when looking at differences with the tower fluxes, so it will not be reflected in the weights. An example could be the ET estimation for those biomes and moments when interception can be a relevant component of the fluxes. If the tower fluxes are not properly capturing this component, the physical methods that try to do so (e.g., GLEAM, PT-JPL, MOD16) may be penalized in the weight derivation, compared with MPI, but their fluxes may be more correct. Of course, the problem is how to merge or validate the ET products away from the tower flux data. Given that DOLCE is a global product with a relatively long time series, at least the annually integrated ET could be compared with basin-integrated differences of precipitation and runoff, which may shed more light into the merits of the individual products, the equally weighted mean product, and DOLCE.

---

## Referee Comment (RC2) · Anonymous Referee #2 · 26 May 2017

Six existing gridded ET products are combined using a weighting approach trained by 159 Fluxnet sites. The method is based on the ensemble weighting and rescaling technique suggested by Bishop and Abramowitz (2013). The technique provides an optimal linear combination of ensemble members by using flux tower observational dataset. I have questions about why do you choose to produce DOLCE at 0.5 deg. resolution? Not 0.25 deg or 0.05 deg (the possibilities provided by MOD16, GLEAM ET), even as high as 1km (the possibility will discussed later in my report). I need a strong rebuttal from the authors here. In addition, before producing DOLCE, are the weighting ETs resampled to 0.5 degree? Or how did you combine ET at different resolution into 0.5 degree DOLCE? My understanding is that ET product at either 0.5,

or 1 deg., most of the flux site grid is in-homogeneity, thinking about the land surface covers can varies a lot in 50km*50km resolution. The more higher resolution of ET, the more higher chance could the site measurement represent information for a gird. Please check the reference: (Anderson et al., 2012).

DOLCE in this work does not make an extension for spatial resolution, or temporal resolution, even time coverage (10 years, 2000-2009). It is not encouraged for publication, which is similar to already published by Mueller et al. 2015. However, the method used to merge ET products is very useful. But the way of using it at 0.5 deg is not an optimal one.

I would say whether the site is reported homogeneous in the evaluations is related to the scale. To asses site homogeneity by ET producer is not fare for the operation of these flux sites. I understand site PIs would seek to ensure the site represent one typical land surface. All the flux tower can be taken as homogeneous sites when the evaluated ET is at 10 meter resolution. Move to 0.5 deg grid, all the flux is located in an in-homogeneous grid. Thus you cannot say 'homogeneous sites', but 'homogeneous site grid'. Or most likely homogeneous due to good matching between 0.5 deg grid and flux site. This also intrigue my interests, if you use the method to combine and calibrate existing ET to be a ET at 1km or higher resolution, which makes all available flux sites homogenous at this scale and can be used to calibrate weighting ET, more land covers can also be used not only DOLCE Tier 1, 2 and 3, say as IGBP 18 land covers. Why don't you use IGBP land covers to replace tier 1, 2 and 3 to derive weight for the 18 land covers? More classification or tiers can also help you produce DOLCE at higher resolution. You may say weighting ET cannot provide information at lower resolution. But the weighting ET can be calibrated with flux site at higher spatial resolution. Each weighting ET can be resampled and calibrated to e.g. 1km resolution. Then you can further combine them into a DOLCE high resolution ET. This work will make the DOLCE more useful. The current 0.5 deg. ET does not differentiate from other fused/merged ET at coarse resolution, e.g. Landflux.

In addition, I also think about if your collection of flux tower data is not enough, which may lead to an biased weight for DOLCE calculation. One reviewer also pointed out the limitation of flux tower for the tropical region. Especially, the flux tower play an important role in your method. You mentioned that 'irrigated sites' was excluded. What's the purpose of doing this? is it believed that all the weighting ET product cannot estimate ET for irrigated crops? If yes, this means DOLCE can also has a big errors for irrigated regions or human influenced regions. Then this need pointed out or at lease add discussion about shortage of this dataset. If no, I would be interested to look at the performance of weighting ET products and DOLCE at irrigated flux sites, since irrigated crops may also influence global water balance. We cannot blind to this issues when producing a global ET. Can't we? I agree if you remove irrigated sites or HT sites, this will makes your ET or paper looks better, but in reality it also expose the shortage of the method and products.

More specific comments list below.

There are a lot of errors in the Table 1. This has been pointed out by Carlos Jimenez. In addition, I also found some ET not included. This is also a 0.05 deg. monthly ET. You may find here: http://en.tpedatabase.cn/portal/MetaDataInfo.jsp?MetaDataId=249454

Table3. Weighting ET products have negative or positive mean bias, if you give 0.5 weight to MPIBGC (3.837 mean bias) and GLEAM-v2B (-3.571) respectively, then DOLCE tier 1 will have a mean bias of 0.266 W/m^2. I also calculate mean bias with the weight and mean bias in table 3 for tier 1, then I get a mean bias of 0.9021 (0.041*3.756+0.495*3.837+(-0.026*6.180+....=0.9021), why do you think the weight provided in table 3 is better than my suggested weight of the two ETs?

Page 11 line28-30, please see my comments above

Page 5, at what time resolution did you do the energy balance correction in the two equations or methods? Monthly or half-hourly? I would not expect 'no qualitative differences' between bowen and residual term correction at half-hourly flux data. Secondly,

most of the flux sites have no ground heat flux but soil heat flux at 5cm. What's your consideration of G used in the correction.

Fig.4 LFD or LDF*,LFA or LFA*?

Figure 7, there is no values for the Sahel desert. This is due to non-values from DOLCE or LandFlux? Then I found DOLCE has ET estimate for Sahel desert in Fig. 8. Please explain it.

Fig. 8. How did you say reliability of uncertainty is low? DOLCE has uncertainty with monthly temporal resolution, am I right? I have difficulty understanding 'seasonal variability of global mean ET and its associated uncertainty'. It's better say 'spatial distribution of a) global ET and (b) its associated uncertainty (standard deviation)in Winter and Summer,'.

Line 10. 'Together with the reasonable density .. are reasonably well constrained.' need rephrase.

Page 1, Line 16, 'point-based estimates of flux towers provide information at the grid scale of these products.', are you sure point flux tower can provide information at 50*50 km pixel? Please check my comments above.

Page 1, Line 21, These ET products differ in their data requirements, the approaches used to derive them and their estimates (Wang and Dickinson, 2012). This is well known. No need citation here.

Line 24, we provide an even stronger vindication of this relationship. Which part show this? Please give some explanation.

Please use either 'Time-space step' or 'Space-time step';

Acknowledgements Where did you use ERA-inerim by saying 'The ERA-Interim reanalysis data are provided by ECMWF and processed by LSCE'.

References: Annan,…..n/a-n/a

Fisher, JB has been listed two times.

Miralles, D.G, also listed two times.

Please check the standard format for references used on HESS website

Table s1, please also add RMSE, correlation and mean bias values for each site. These information is also important for DOLCE dataset users.

Reference: Anderson, M.C., Allen, R.G., Morse, A. and Kustas, W.P., 2012. Use of Landsat thermal imagery in monitoring evapotranspiration and managing water resources. Remote Sensing of Environment, 122: 50-65.

---

## Author Comment (AC1) · 3 Jul 2017

**Manuscript hess-2017-147 entitled "Derived Optimal Linear Combination Evapotranspiration (DOLCE): a global gridded synthesis ET estimate"**

We would like to thank the reviewers for their constructive comments on our manuscript. This document outlines our point-by-point responses to the reviewer #1 comments and the improvements made to the manuscript.

**Response to Reviewer #1**

**General comments**

1. There is obviously a question regarding the representativeness of the FLUXNET stations to their regions (grid box average or dominant surface conditions), which the paper addresses fairly well. However, one specific issue jumps out at me: what about rare representatives, particularly in the tropics? FLUXNET is woefully thin on stations at low latitudes. The out-of-sample tests are like an OSSE, but there is little sampling in the tropics to play with here. What was the specific effect of denying low-latitude stations? Can we use this study to make case to prioritize FLUXNET expansion into the tropics, S/SW Asia and the Southern Hemisphere? I would like to see this issue discussed more, where the global maps are presented and also in the Discussion. Motivating targeted FLUXNET network expansion would be a good "broader impact" of this study.

This is a great point. We have extended the discussion to try to address this:

*There are relatively few towers located in the Southern Hemisphere and the tropics (14 out of 159 sites) and none located in the dry climates over South West Asia and North Africa. The weighting was therefore mostly driven by the ability of products to match sites located in the temperate and cold zones of the Northern Hemisphere, so that performance in climate zones with low FLUXNET site density was under-represented when deriving DOLCE. This might raise questions about the performance of DOLCE in the tropics and the Southern Hemisphere. To evaluate DOLCE in these areas we calculated the four site metrics separately for two groups of sites, 1) those located in the Northern Hemisphere excluding Tropics and 2) sites located in the Tropics and/or Southern Hemisphere. We excluded the two sites ID-Pag and AU-Fog in this exercise since both are wetland sites, and so would complicate a determination of whether these two groups had notable behavioural differences.*

*If systematic behavioural differences did exist between these two groups, we would expect relatively poorer performance of DOLCE at group2 sites compared to group1 sites. The results, shown in Fig. 10, appear inconclusive. DOLCE performed marginally worse at group2 sites overall, however with the limited number of sites in group2, the validation of the performance of DOLCE in the tropics and Southern Hemisphere remains somewhat uncertain. The uneven distribution of eddy-covariance sites between the Northern and Southern Hemisphere and across the different climates might also explain why much of the largest seasonal differences DOLCE-MPI and DOLCE-LANDFLUX-EVAL shown in Fig. 6 and Fig. 7 reside in the low latitudes (tropics) and the Southern Hemisphere and the persistent differences between DOLCE and LANDFLUX-EVAL in the tropics throughout the year. The expansion of the FLUXNET network into these areas that are lacking observations is clearly something that would improve DOLCE and LSM evaluation more broadly.*

2. **What about the uncertainty/error in the FLUXNET observations? They certainly contain random errors and systematic biases. For a measured quantity that has a red-noise spectrum (following a Markov process) the random error can be estimated from the**

**behavior of lagged autocorrelations (cf. http://dx.doi.org/10.1175/JHM-D-15-0196.1). Random error will systematically degrade correlation and affect other statistics (cf. http://dx.doi.org/10.1175/JHM-D-15-0063.1). Systematic error is more difficult to identify, but the energy balance corrections in FLUXNET2015 give some clue (see specific comments below). All these mean that using FLUXNET for cal/val is itself flawed and imperfect. On the other hand, model ET is inherently precise (no random error) and can be used to estimate/differentiate this type of error from others by noting its statistical differences from the instrument records. This issue should be acknowledged and discussed, including how the assumptions in DOLCE regarding errors (that they are uncorrelated) affect results. In other words, more discussion about uncertainties.**

Yes. We have extended the discussion to address this point, at least to some degree:

*Many studies have analysed the systematic and random errors of latent heat flux in FLUXNET measurements (Dirmeyer et al., 2016; Göckede et al., 2008; Richardson et al., 2006). These studies have detected errors of magnitudes that cannot be neglected. A recent study (Cheng et al., 2017) showed that the computed eddy-covariance fluxes have errors in the applied turbulence theory that lead to the underestimation of fluxes, and that this is likely to be one of the causes of the lack of surface energy closure. In this study, we 1) used the flag assigned to the observed flux, to filter out the low quality data and 2) used energy-balance-corrected FLUXNET data which has higher per-site mean values than the raw data at most of the sites (85% of them). We expect that filtering together with the use of corrected data will reduce the magnitude of the uncertainty in the observational data used here and compensate to a certain extent for the underestimation due to the systematic errors. However, we have not formally explored a range of approaches to addressing this. The possibility of systematic biases in FLUXNET data remains, and this could clearly lead to systematic biases in DOLCE.*

*We have also assumed that error across sites is uncorrelated, which, given the distribution of sites, is unlikely to be true, meaning that the effective number of sites is probably somewhat smaller than those shown in Figure 2. Given this dependence is likely to vary depending on a*

*range of time varying factors, we have left the job of attempting to disentangle this issue for future work.*

3. **In the Supplement: There are captions for Fig S1 and Fig S2, but the PDF does not contain any figures!! Please recheck the rendering.**

   We thank the reviewer for spotting this, we now added the missing Figures in the Supplement.

**Specific comments:**

4. **P3 L6: Change "85 FLUXNET tower data" to "85 FLUXNET towers"**

   We've corrected this in the manuscript.

5. **P3 L8: "ground-truthed" is not an appropriate verbification for this context. Change '...gridded ET data sets are "ground-truthed" using flux tower data from FLUXNET...' to '...tower data from FLUXNET provide ground truth for gridded ET data sets...'**

   Thanks for picking this up, we've made the change.

6. **P5 L16: Change "where" to "were"**

   Thanks for picking this up, we've corrected this in the manuscript.

7. **P7 L22: RSD cannot convey whether the variance is too large or too small. So for the changes shown in Fig 3b - we cannot tell whether the improvement is due to an increase or decrease in standard deviation. Furthermore, what about locations where mean ET (denominator) is near zero - does that explain some of the very large values and changes?**

   The reviewer is right; as a result of this comment and a comment by Carlos Jimenez, we've replaced the relative standard deviation metric (RSD) with a modified relative standard

deviation MRSD defined as $\frac{\sigma_{\text{dataset or observation}}}{max(mean(observation),\ q)}$. This addresses the two issues and removes the potential for improvement in the RSD metric simply because the mean has improved. We have also added a fourth panel to this Figure showing improvement in the mean, for reference. We added the text below to explain the new metric:

*We use a modified relative standard deviation metric MRSD that measures the variability of latent heat flux relative to the mean of the flux measured at each site. This ensures that a comparison between MRSD for a product and observations can tell us whether a product's variability is too large or too small (unlike relative standard deviation). The term 'q' is a threshold representing the 2$^{nd}$ percentile of the distribution of observed mean flux (i.e. temporal mean ET) across all sites (about 13 W/m2), which guarantees that MRSD calculated across many sites is not dominated by sites where the mean flux (denominator in MRSD Equation above) approaches zero. We looked at the bias in MRSD for each product considered- i.e. |MRSD$_{dataset}$ - MRSD$_{observation}$|, and showed the performance improvement of the weighted mean.*

8. **Fig 3: So the spread is across 5000, and each of those is an average of 25% out-of sample stations, right? It is not 5000x172x0.25 points. Please make clear.**

The reviewer is right. We now clarified this in the figure caption:

*Figure 3: Box and whisker plots displaying the percentage improvement that the weighted product provides in the 25% out-of-sample sites test for four metrics: MSE (a), MRSD (b), COR (c) and Mean bias (d), when compared to equally weighted mean of the Diagnostic Ensemble (Dmean), aggregated Reference Ensemble (Ragg) and each member of the reference ensemble. Box and whisker plots represents 5000 entries, each entry is generated through randomly selecting 25% of sites to be out sample*

9. **Also, Fig 4 suggests individual o-o-s stations frequently fare worse. Transferability of calibrations appears to be kind of weak, which is not a surprise. Calibration transferability is a very difficult enterprise. This should also be acknowledged, either here or in the**

**Discussion section (wishing PILPS-San Pedro had been completed as it would have really shone a light on this problem).**

Yes, this is indeed worth mentioning. We have modified the results section to read:

*In both cases it is important to note that many individual sites agree poorly with the weighted product compared to some other products. The distinction between the results shown in Fig. 3 versus Fig. 4 serve to highlight that DOLCE, and indeed any other large scale gridded ET product, is not suitable for estimation of an individual site's fluxes, even if prediction over many sites shows notable improvement.*

10. **P9 and Fig 4: "widespread out-of-sample improvement that this approach offers over existing gridded ET products" seems like a bit of an overstatement - there are frequently locations that fare worse, and sometimes much worse. In Fig 4, there is typically a tremendous range, and usually the central two quartiles encompass the zero line. For RMSE it appears that _20-49% of the time the estimates are worse than other products and methods, 30-55% for RSD, and 20-55% for COR. While there is definitely a net (and welcome) improvement in almost all cases, often the DOLCE estimate is worse. This should be acknowledged.**

This is indeed a fair criticism. We have modified this to read

"*On the basis of the aggregate out-of-sample improvement that this approach offers over existing gridded ET products*".

11. **P9 L26: "Standard Deviation (SD) difference" - this is clearly not the same as RSD defined earlier - please define, which minus which?**

The reviewer is right, this is not the same as RSD. We've clarified this in the manuscript:

Standard Deviation (SD) difference (i.e. $\sigma_{DOLCE} - \sigma_{observation}$)

**12. Table 2 has little accompanying discussion, and what is there is very shallow adding little to comprehension. Please discuss more or remove the Table if it does not warrant discussion.**

We thank the reviewer for his suggestion, we removed this table from the manuscript.

**13. Fig 6: Here it could be said you use DOLCE to estimate MPI, and most places have a positive bias. But the energy-balance-corrected fluxnet data, which close the surface energy balance by construct conserving Bowen ratio, consistently increases ET compared to the raw measurements (at 107 of 122 FLUXNET2015 Tier-1 stations during JJA by my quick calculation). There is recent independent indication that tower sites bias low because of errors in the turbulence theory applied to estimate fluxes (see: http://dx.doi.org/10.1002/2017GL073499). Could FLUXNET instrument error (and the simplicity of the energy closure correction) contribute to systematic biases in DOLCE? Please discuss here or in the Discussion section.**

Yes, of course it could. We have added to the discussion to make this clearer:

*Many studies have analysed the systematic and random errors of latent heat flux in FLUXNET measurements (Dirmeyer et al., 2016; Göckede et al., 2008; Richardson et al., 2006). These studies have detected errors of magnitudes that cannot be neglected. A recent study (Cheng et al., 2017) showed that the computed eddy-covariance fluxes have errors in the applied turbulence theory that lead to the underestimation of fluxes, and that this is likely to be one of the causes of the lack of surface energy closure. In this study, we 1) used the flag assigned to the observed flux, to filter out the low quality data and 2) used energy-balance-corrected FLUXNET data which has higher per-site mean values than the raw data at most of the sites (85% of them). We expect that filtering together with the use of corrected data will reduce the magnitude of the uncertainty in the observational data used here and compensate to a certain extent for the underestimation due to the systematic errors. However, we have not formally explored a range of approaches to addressing this. The possibility of systematic biases in FLUXNET data remains, and this could clearly lead to systematic biases in DOLCE.*

*We have also assumed that error across sites is uncorrelated, which, given the distribution of sites, is unlikely to be true, meaning that the effective number of sites is probably somewhat smaller than those shown in Fig. 2. Given this dependence is likely to vary depending on a range of time varying factors, we have left the job of attempting to disentangle this issue for future work.*

14. **Re Fig 7: Much of the largest differences are at low latitudes where there is little FLUXNET data for calibration. Please discuss, as I see this as a major issue (more with FLUXNET station distribution than your methods, but the problem of representative ET estimates in the tropics is an ongoing concern).**

    We have addressed this issue in our response to the first point raised above.

15. **P10 L21: "ET doesn't exhibit any seasonal change over Greenland and the deserts in North Africa..." - not expected in the absolute, because mean values are tiny. It would be more informative to also show relative (percentage) changes, which are more relevant for local water balances.**

    A good point. We have now added two extra plot in Fig.8 that show the seasonal variability of 1) ET estimates and 2) uncertainty estimates, we changed the plot titles, made the caption clearer:

*Figure 8: Seasonal (a) global mean ET and (b) its variability (standard deviation), (c) time average of uncertainty (the standard deviation uncertainty shown in Equation 7) (d) standard deviation of uncertainty over time (e) reliability, defined as high ($\frac{Uncertainty\ SD}{mean\ ET} \leq 1$ in blue), medium($|mean\ ET| \leq 5$, $Uncertainty\ SD < 10$ and $\frac{Uncertainty\ SD}{mean\ ET} \geq 1$ in green) and low (in red). DJF is shown in the left column and JJA in the right column.*

and commented on the plots in the text:

*The spatial distribution of DOLCE mean ET and its seasonal variability (standard deviation) over the austral Summer (Dec–Feb) and Winter (Jun–Aug) from 2000 to 2009 is shown in Fig. 8 (a) and (b) respectively. The seasonal variability of ET is larger in the warm season but is always small over Antarctica, Greenland and the deserts in North Africa (Sahara), the middle east (Arabian Peninsula desert) and Asia (i.e. Gobi, Takla Makan and Thar). The average uncertainty shown in if Fig. 8 (c) is bigger in the warm season, this is in agreement with the relatively large size of the flux in the warm season, and its seasonal variability shown in Fig. 8 (d) is also in agreement with the seasonal variability of the flux.*

16. **Discussion §: Please also speculate whether some spatial variability in weighting could improve estimates further, even if only in 2 or 3 categories of weights.**

    We agree this is worth exploring. That's why we performed clustered weighting where each cluster had a different set of weights. We accept this might not have been clear enough in the manuscript, we clarified this in the discussion:

    *In this study, we sought a single weight for each product to apply globally. But we have a reason to believe that different products are likely to perform better in different environments, so that different weights in different climatic circumstances might well improve the result of weighting overall. A similar suggestion was made in the studies of (Ershadi et al., 2014) and (Michel et al., 2016) who highlighted the need to develop a composite model, where individual models are assigned weights based on their performance across particular biome types and climate zones. We therefore tried to cluster flux tower sites into groups (such as vegetation type) so that each group maintains enough members to allow the in- and out-of-sample testing approach used above. We tried clustering by vegetation type, climate zone and aridity index, and implemented the same one site out-of-sample testing approach as above, but this time, in each cluster different sets of weights will be assigned to the weighting products.*

17. **Fig 9: Only stations in the tropics are 2 EBF stations; the savannah stations are in southern Africa. These are not o-o-s results, right? I would have expected these types to**

**stand out more, but perhaps the samples are biased to the extratropical stations in the categories. Thoughts?**

We have added this point to the discussion:

*The EBF box and whisker plot in Fig. 9 (d) shows the correlation of DOLCE at eight EBF sites, out of which two sites are located in the tropics. The lowest correlation seen in this biome type is at the tropical sites ID-Pag (0.26) and BR-Sa3 (0.62). This suggests that DOLCE tends to represent ET at the extratropical sites better than the tropics, and this is not surprising since most of the sites that were used to calibrate DOLCE were extratropical sites.*

**18. Also Fig 9: Crops are tricky. The category is a catchall that is unsatisfactory because there is such variability in phenology, seasonality, stomatal resistances, etc. Are any of the CRO stations rice (which acts very different because of seasonal flooding). Not surprising many of the biggest errors are there.**

Good point. In the site description provided by FLUXNET the crop type is not always clear. There is only one rice site that we are sure about, but we excluded this site from the weighting because it was found in the list of irrigated sites. We excluded all irrigated sites. We've added the text below to explain the reasons:

*In (6), we expect that some of the weighting models will largely underestimate the flux at irrigated sites, a result of a missing irrigation module in their scheme (Miralles, 2011; Jung, 2011). Because of this, the error bias of these models at the irrigated sites will modify the mean error bias (i.e. mean bias across all the sites) significantly, which will affect the weighting in favour of the products that can represent better irrigation. We excluded these sites as we do not want the products to be weighted for their inclusion/non-inclusion of physical processes.*

We expanded our analysis as suggested by reviewer #2 point 10 to test the performance of DOLCE in three irrigated sites, we've added the text below to show this:

*We tested the performance of DOLCE at three irrigated sites that were excluded from the weighting, for reasons explained earlier by computing the four statistics. A description of these sites and the results are shown in Table 2. The results show that the performance of DOLCE is reasonable at US-Ne1 and US-Ne2 and low at US-Twt. These results are discussed further below.*

*DOLCE has also shown a weak performance at US-Twt, which is an irrigated rice paddy. This site gets flooded in spring and drains in early fall, then the rice is harvested. Only 9 months were available for this site, which coincide with the flood and drain period between spring and fall. DOLCE could not depict the flooding and draining event, probably because none of the weighting products can represent such phenomena, so it is expected that the effects of seasonal flooding are not represented in DOLCE.*

19. **P13 L1: "...for example anthropogenic water management..." - but it was stated earlier that irrigated sites were excluded. Please expand on this comment.**

    We addressed this concern in the previous point.

20. **Conclusion: Please also state plans for updates, future versions, perhaps (hopefully) covering a longer time period!?**

    We thank the reviewer for his suggestion. We have added future improvement of DOLCE in the conclusion.

21. **References: There seem to be a lot of redundancies in author lists for papers with names showing up 2 or 3 times for the same entry. Please check.**

    Thanks for picking this up, we have now removed the redundancies and made the appropriate corrections.

---

## Author Comment (AC2) · 3 Jul 2017

**Manuscript hess-2017-147 entitled "Derived Optimal Linear Combination Evapotranspiration (DOLCE): a global gridded synthesis ET estimate"**

We would like to thank Carlos Jimenez for his constructive comments on our manuscript. This document outlines our responses to his comments and the improvements made to the manuscript.

**Response to Short Comments**

**Product Selection**

While acknowledging the difficulties of finding the rights products to derive a synthesis product, I am a bit surprised about some of the choices.
As the authors state, product selection should follow the criteria of product diversity, so ideally single algorithms with different strengths and weaknesses should be combined together. In that sense, I would have not considered an already synthesis product such as LandFlux-Eval as a possible candidate for the merge.
In my opinion, a more valid alternative regarding LandFlux efforts could have been the three single products publicly available based on different ET algorithms (https://hydrology.kaust.edu.sa/Pages/GEWEX_Landflux.aspx). Also, I do not see much interest in combining obsolete versions of products with the current, and presumably better, product, as it has been done for GLEAM. Just GLEAM V3A (and perhaps GLEAM V3B) seems to me a better option. An interesting product is MPI. This is a global extrapolation of the tower fluxes, the same tower fluxes that are used to decide on the merging weights, and quite different to the other products, which we could consider more "physically" based and less "calibrated" with the tower data. It is not surprise then that MPI is by a large margin the more weighted product. I do not deny that it can be a valid product for the merge, although much less independent than the other products with respect to the tower data. In that sense, it could have been very

**informative also to see how the merging works, and how the weights are distributed, when that product is left out**

Yes, this is indeed a reasonable question. One key distinction of the weighting approach here is that it accounts not only for the performance differences between products, but also the error covariance between them (as noted in Section 2.3). So, if a product were added that were a near copy of another, it would not degrade the performance of the weighting at all. While variants of GLEAM are indeed likely to be similar, small structural differences might mean that there is in fact an advantage to using a nominally inferior version in addition to the latest version. This might explain why GLEAM-V2A was assigned a negative coefficient when the three GLEAM products participated in the weighting (tier1). By assigning a negative weight, the weighting was removing redundancies or data that was not adding any information to the weighting.  It was not possible to include GLEAM v3B in the current version, due to the limitation in the covered time period which doesn't include 2000-2003.

Testing how the weighting would perform *without* MPI is an interesting idea. We therefore did exactly that - removed MPI from the weighting as a separate experiment and included this in the manuscript. Performance, perhaps surprisingly, was very similar:

*it is not surprising that MPIBGC was the most weighted product since it is highly calibrated with flux tower data. In a further analysis, we left MPIBGC out and we performed the out of sample tests using the five remaining products. We wanted here to test how the weighting will perform without MPIBGC.  The plots in Fig. 11 and 12 show the results of 25 % out-of-sample test and one site out-of-sample test respectively. Overall, the weighting offers a smaller performance improvement than that offered when MPIBGC is a member of weighting ensemble (Fig. 3 and Fig. 4 (a-d)). The distribution of the weights when MPIBGC is absent from the weighting is 0.3 for both PML and GLEAM_v3A, o.2 for GLEAM_v2B, 0.13 for MOD16 and 0.07 for GLEAM_v2A.*

The new plots Fig.11 and 12 are shown below

[Figure]

*Figure 11: Box and whisker plots displaying the percentage improvement that the weighted product excluding MPIBGC provides in the 25% out-of-sample sites test for four metrics: MSE (a), MRSD (b), COR (c) and Mean Bias (d), when compared to equally weighted mean (Dmean) of the Diagnostic Ensemble, aggregated Reference Ensemble (Ragg) and each member of the reference ensemble. Box and whisker plots represents 5000 entries, each entry is generated through randomly selecting 25% of sites to be out sample*

[Figure]

*Figure 12: Box and whisker plots displaying the percentage improvement that the weighted product excluding MPIBGC provides in the one out-of-sample sites test for four metrics: MSE (a), MRSD (b), COR (c) and Mean Bias (d), when compared to equally weighted mean (Dmean) of the Diagnostic Ensemble, aggregated Reference Ensemble (Ragg) and each member of the reference ensemble. Products marked with * have limited spatiotemporal availability relative to the diagnostic ensemble, and testing against the LFA, LFD, CS and PT products was limited to 110, 108, 108 and 72 sites respectively.*

**Spatial and temporal resolution**

**The resolutions given in the Table 1 seem wrong for a number of products (e.g. MODIS original resolution is 1 km, Zhang 2010 is 0.05 deg, MPI and Zhang 2015 0.5 deg). Also, the periods of available data should also be revised for GLEAM (I think GLEAM V2A covered 1980-2011 and V3A 1980-2014).**
Thanks for spotting this, we have corrected the errors in Table 1

**As far as I can see, the only datasets limiting the study period to 2000 is MOD16 and GLEAM V2B. Perhaps the products going into merging could have been separated in two groups, a bit similar to what has been done regarding geographical coverage: from a much earlier year than 2000 will all but MODIS and GLEAM V2B, and from 2000 including MODIS and GLEAM V2B. That would have resulted in a much longer DOLCE dataset, presumably based on a larger collection of tower data and more ET products. A shorter time period than monthly will result is a more useful product. Daily will a be a better objective for future developments, although it will require a different selection of products, possibly more based on the "physically" based diagnostic ET products, where daily is a common time scale. I would suspect a more complex merging exercise, given the larger amount of time variability that needs to be captured by the merge product ET and FluxNet datasets**

Great comments and suggestions. In this work, limiting DOLCE to be derived from diagnostic (as opposed to overtly model-based) products only made the choices of temporal and special resolution limited. As suggested, extending the period of DOLCE by using different products for different time slices of DOLCE is a great idea, and will likely be an objective for further development in future versions of DOLCE. Applying the same merging technique on daily diagnostic ET products requires more complex analysis at both the site scale (observation) and grid scale (gridded products), although this is worth investigating for future versions. Thanks!

**Tier 3 (i.e, Greenland and Antarctica) is just a very close weighting of an obsolete version of GLEAM and a newer version. It seems a bit awkward to distribute that as**

**part of a synthesis product. It may have been better to just remove those regions from the synthesis product given that nearly no one dares to estimate ET over there (understandably).**

Yes, agreed. We however wanted to produce DOLCE with global coverage (requiring tier1, tier2 and tier3), and did make an effort to show the level of reliability of each tier in Fig. 8. We published tier1, tier2 and tier3 in separate files in order to give the users of DOLCE the flexibility of selecting the most appropriate product.

**Some of the ET datasets considered are based on algorithms that estimate separately interception, evaporation from the canopy, and evaporation from the soil (e.g, MOD16, PT-JPL, and GLEAM). Under the assumption that routine EC observations perform very poorly for rainy conditions, in principle interception is not captured by the tower observations. In some recent ET evaluations of these products the tower data has been filtered to remove rainy periods and the interception component has not been evaluated. A discussion about this could have been interesting, given that, as far as I can see, the tower data is not filtered for precipitation conditions, and the merged product is a total ET product.**

Yes, we haven't filtered the sites for rainy periods, especially since we're working with monthly data. Something to explore in the future, especially if we investigate using daily data.

**Regarding the energy closure issue, the text may give the impression that fluxes correction is always possible, but a large number of stations do not measure Rn and/or G. Given that only corrected fluxes are used in the study, I imagine that a number of stations have to be discarded as they corrected fluxes were not available (FLUXNET-2015) or could not be estimated (LaThuille-2007). This may be worth mentioning**

We only corrected LaThuile sites for energy imbalance at sites that had measurements of all the component fluxes. We applied quality control and filtering for G, H and Rn as

highlighted in steps (2) and (3), section 2.2. This is perhaps not clear enough in the manuscript, we therefore added the text below to clarify this point:

*Applying a correction technique for energy imbalance at LaThuile sites required applying (2) and (3) for the other components of energy imbalance (i.e. $R_n$, G and H), which means that the sites that had to undergo a correction for the energy imbalance, should have monthly estimates for all the fluxes of the energy budgets, where each monthly value has been calculated from at least 15 daily mean flux values. Because of this constraint, many sites were disregarded from the analysis.*

**Merging technique**

**If I understand the method correctly, the weights are global (i.e. one value for all pixels), time-invariant (i.e., an annual value), and the bias-correction is what Bishop, 2013 calls a "global bias" correction (i.e., is a single value per product using all towers in the in-sample training dataset). If this is true, if Product A was performing better than product B over some biomes and/or at some periods, the method cannot be used to weight more or less the products to reflect that difference in performance. If I am correct, I wonder if there is a way to modify the weighting to take into account those differences. We typically see that ET products perform differently at different biomes and/or seasons, so it may be advantageous to capture this in the weighting. Bishop, 2013 was quite illustrative about this. Given that for that temperature example the "truth" was quasi-global (i.e., not over a very few pixels like for the ET), just a "per-cell bias" correction, even without weighting, outperformed the weighted product with a prior "global bias" correction. Of course, the "per-cell bias" correction and weights cannot be be applied here, given the limited geographical coverage of the flux "truth", so the problem is more complex.**

**I guess your first take at this is your sites clustering based on vegetation type, presented in some detail in the Supplement. You conclude that it did not improve overall the DOLCE performance based on a global analysis, but I would be interested to see the results (in the Supplement I can download the figures summarizing the**

**analysis are missing). Perhaps other schemes that better cluster the flux behavior are worth investigating for future versions of DOLCE**

Yes, we calculated a single bias term, and assigned one weight to each ET product to be applied globally. As you also note, we did apply clustered weighting, where we derived cluster dependent sets of bias terms and weights for weighting the ET products. We tried clustering by 1) vegetation type, and also 2) climate zone and 3) aridity index (2 and 3 not shown in this study), and we implemented the same one site out-of-sample test, but this time separately in each cluster. We calculated different sets of weights for the ET products based on their performance differences in the different clusters, but none of the clustering succeeded in deriving a better weighted product overall. We added the text below to clarify the clustered weighting.

We show the results of one site out-of-sample test where different weights were assigned to products at each biome type (in supplementary material). We apologize for not putting the figure in the supplementary material, this has now been added.

*In this study, we sought a single weight for each product to apply globally. But we have reason to believe that different products are likely to perform better in different environments, so that different weights in different climatic circumstances might well improve the result of weighting overall. A similar suggestion was made in the studies of Ershadi et al. (2014) and Michel et al. (2016) who highlighted the need to develop a composite model where individual models are assigned weights based on their performance across particular biome types and climate zones. We therefore tried to cluster flux tower sites into groups (such as vegetation type) so that each group maintains enough members to allow the in- and out-of-sample testing approach used above. We tried clustering by vegetation type, climate zone and aridity index, and implemented the same one site out-of-sample testing approach as above with different sets of weights in each cluster.*

**Results**

**It is stated that the MSE plot in Fig. 3a shows the MSE of the weighted product being better than the ensemble mean, but I do not see it. The central line of the first whisker**

**box in Fig 3a is at the zero line. Perhaps I am missing something regarding how to read these plots. As a side note, it may be good to say what the end of the whiskers represents. In most occasions it is used to represent the max and min of the data, but it is not always the case.**

We have modified both the plots and their description in the manuscript. As noted in our response to Reviewer 1, we have modified the RSD metric and added the performance of DOLCE on mean values as well in a fourth panel. We have also clarified the description of the box plots as follows:

*We display the results of performance improvement datasets calculated in (a-c) above in 12 box and whisker plots. In each boxplot, the lower and upper hinges represent the first ($Q_1$) and third ($Q_3$) quartiles respectively of the performance improvement datasets, and the line located inside the boxplot represents the median value. The extreme of the lower whisker represents the minimum of 1) max(dataset) and 2) ($Q_3$ + IQR), while the extreme of the upper whisker is the maximum of 1) min(dataset) and 2) ($Q_3$+ IQR)), where IQR is the interquartile range of the performance improvement dataset. If the median performance improvement is positive, this indicates that the weighting offers an improvement in more than half of the data presented by the boxplot.*

Note that (a-c) are explained in section 2.4

**In the same Fig. 3a, the whisker boxes for the individual products make me think again about the MPI product. Based on the "heavy" calibration of MPI with the same tower data used to derive the weights, I would speculate that if the MPI product was removed from the merging, the percentage improvements of this new weighted product (i.e., without MPI) over the individual products will be much smaller. This may give a different perspective of the exercise regarding the skill of the tower-based merging to combine the more "tower-independent" physically based products.**

Please see our response to a similar point above earlier in this document (in Product selection).

**If we just concentrate on the improvements of the weighted product with respect to the equally weighted product (i.e., first whisker box in Fig 3 a-b-c), the gain in performance of the weighted product seems small. Again, if I read these plots correctly, the gain for MSE and COR is minimal, only the RSD shows some improvement. But given the definition of the RSD metric, I wonder is this is mostly associated to the bias correction. If I understand this correctly, after the bias correction mean-dataset and mean-observation will be equal, so RSD is abs(sigma-dataset –sigma-observation). In other words, I am wondering about a comparison of the equally weighted product, but with a bias-correction first, and the weighted product. I am assuming here that the equally weighted mean did not involve a prior bias correction, as nothing was stated in the paper, but I may be wrong.**

Yes, the plots in Fig 3 a,b and c show that overall, the improvement of the weighted product is noticeable with respect to the reference products and marginal with respect to the equally weighted mean. As a result of this comment and those by Reviewer 1, we've replaced the relative standard deviation metric (RSD) with a modified relative standard deviation MRSD defined as $\frac{\sigma_{\text{dataset or observation}}}{max(mean(observation),\ q)}$. This removes the potential for improvement in the RSD metric simply because the mean has improved. We have also added a fourth panel to this Figure showing improvement in the mean, for reference. We added the text below to explain the new metric:

*We use a modified relative standard deviation metric MRSD that measures the variability of latent heat flux relative to the mean of the flux measured at each site. This ensures that a comparison between MRSD for a product and observations can tell us whether a product's variability is too large or too small (unlike relative standard deviation). The term 'q' is a threshold representing the $2^{nd}$ percentile of the distribution of observed mean flux (i.e. temporal mean ET) across all sites (about 13 W/m2), which guarantees that MRSD calculated across many sites is not dominated by sites where the mean flux (denominator in MRSD Equation above) approaches zero. We looked at the bias in MRSD for each product considered- i.e. $|MRSD_{dataset} - MRSD_{observation}|$, and showed the performance improvement of the weighted mean.*

It is also true that the equally weighted mean used here doesn't involve any bias correction. Of course, some of the improvement offered by the weighting is owing to the bias correction and some comes from the weighting. We have now separated these effects in Figure S3 in the supplementary material, and referred to it in the results section:

*Part of the success of the weighting approach relative to the multi-product mean is due to the bias correction applied before the weighting. Figure S3 in supplementary material separates the effect of each step.*

**In the HOM and HET comparison, I see very small improvements in MSE, larger for RSD, and not much for COR (the median of the whisker box is for MSE and COR is at the zero percentage line). And I wonder if the separation into HOM and HET sites may have also implied a separation in land covers, so the improvements we see are more related to the weighted product working better for some specific biomes. One may think that land covers such as forested areas are more likely to be represented in the HOM class, compared with e.g. croplands. I wonder if this has been checked, i.e., that the biome representation in HOM and HET classes does not change too much.**

Yes, the weighting calibrated by HOM sites offers only a marginal improvement over the weighing using all the sites (HOM and HET). We looked at the separation of Biomes in HOM and HET cases and we found a clear separation of land covers and their distribution across the HOM and HET case:

1) cropland constitutes more than 20% of the HOM-case sites and 7.6% of the HET-case sites,

2) about 23% of HOM-case sites are forests (EBF, DBF and MF), while more than half of the HET-case sites are forests, the big number of forest sites in the HET case is because most of the MF sites are located on grid boxes identified as EBF and DBF.

3) WET and WSA sites are found only in the HET-case

We added table S2 below in the supplementary material and we modified the text.

*Table S2: Distribution by land cover of HOM-case sites and HET-case sites at both the site scale and grid cell scale*

| Land Cover | HOM-case | HET-case (site) | HET case (grid cell) |
|---|---|---|---|
| CRO | 10 | 7 | 20 |
| CSH | 0 | 1 | 0 |
| DBF | 1 | 16 | 0 |
| EBF | 3 | 5 | 0 |
| ENF | 6 | 22 | 2 |
| GRA | 13 | 27 | 5 |
| MF | 7 | 3 | 32 |
| OSH | 2 | 1 | 4 |
| SAV | 3 | 1 | 6 |
| VEG | 1 |  | 0 |
| WET |  | 5 | 0 |
| WSA |  | 4 | 9 |
| Wa (Water) |  |  | 1 |
| URB (Urban) |  |  | 1 |

*These results nevertheless lead us to expect that if we construct DOLCE by incorporating HOM-case sites only, we might get a better product, but the small number of sites satisfying this property, the fact that the separation of sites into HOM-case and HET-case can lead to a separation of land covers, and the difficulty in defining a meaningful definition for expected flux homogeneity are limiting factors.*

**Regarding the boxplots of Figure 5, it is true that the end of the whiskers are larger for the HET sites, but the RMSE and correlation median and percentiles look slightly better for the HET class. I think this is also worth discussing, as it can potentially indicate again that the differences in performance between HOM and HET classes are small, with just a few HET sites having bad statistics. Given that this is based on one site out-of-sample, I wonder if the bad performance at some individual sites may be nothing to do with homogeneity, but with the fact that the site is one of a kind, so any weights derived from a sample without that site are not informative. Given the location of some the sites, it would not be a surprise.**

An interesting point. It could certainly be the case that the reason for some sites in the HET group showing poor performance of DOLCE might simply be about measurement quality or site uniqueness. The box plots in Figure 5 show the performance of DOLCE in the HOM and HET cases separately. Yes, while the whiskers are larger for the HET sites but we cannot infer from these plots that DOLCE performs better in any of the two groups of sites. We modified the text in the Results and Discussion to make this point clear.

*There is some expectation that the weighting will show better performance if it is trained with HOM-case sites only, although HOM-case sites consist of about one-thirds of the total number of sites used in this study.*

And

*Determining whether DOLCE performs better at HOM-case sites or HET-case sites is inconclusive. Even that the worst performance of DOLCE was achieved in HET-case sites, the boxplots in Fig. 5 (a), (d) show that the value of the median, lower and upper quartiles are better in HET-case for two metrics (i.e. RMSE and COR). While we expect that calibrating the weighting with HOM-case could lead to a better product, we don't expect to see DOLCE overall performing better in any of the two groups.*

**We might expect that (3) would decrease the overall performance of the HET-case sites (see Fig.3). On the other hand, it is very likely that(1) and (2) increase the performance of HET-case sites and eventually compensate the decrease of performance due the WET sites.**

These results nevertheless lead us to expect that if we construct DOLCE by incorporating HOM-case sites only, we might get a better product, but the small number of sites satisfying this property, the fact that the separation of sites into HOM-case and HET-case can lead to a separation of land covers, and the difficulty in defining a meaningful definition for expected flux homogeneity are limiting factors. Determining whether DOLCE performs better at HOM-case sites or HET-case sites is inconclusive. Even that the worst performance of DOLCE was achieved in HET-case sites, the boxplots in Fig. 5 (a), (d)

show the value of the median, lower and upper quartiles are better in HET-case for two metrics (i.e. RMSE and COR). While we expect that calibrating the weighting with HOM-case could lead to a better product, we don't expect to see DOLCE overall performing better in any of the two groups.

**MPIBGC is the larger contributor to DOLCE, with a weight close to ~0.5. Perhaps it is not surprising than the differences of MPI with DOLCE sown in Figure 6 are smaller than with LandFlux-EVAL-Diag in Figure 7, which is not part of DOLCE. I acknowledge that it is quite difficult to come out with the right classes to try to illustrate the reliability of the weighted product. But when I look at figure 8-c what I broadly see is that arid places, Greenland and Antarctica have low reliability, snowed places medium, and the rest high. Perhaps just the uncertainty over the mean ET would have been more informative.**

Great suggestion. We've expanded the plot and showed the seasonal variability of 1) ET estimates and 2) uncertainty estimates, and we've changed the plot titles and rewritten the caption to read:

*Figure 8: Seasonal (a) global mean ET and (b) its variability (standard deviation), (c) time average of uncertainty (the standard deviation uncertainty shown in Equation 7) (d) standard deviation of uncertainty over time (e) reliability, defined as high ($\frac{Uncertainty\ SD}{mean\ ET} \leq 1$ in blue), medium($|mean\ ET| \leq 5$, $Uncertainty\ SD < 10$ and $\frac{Uncertainty\ SD}{mean\ ET} \geq 1$ in green) and low (in red). DJF is shown in the left column and JJA in the right column.*

**Discussion**
**The discussion about a possible selection of HOM sites to construct DOLCE is quite appropriate. Being very strict, and considering that the typical tower fetch is of the order of hundreds of meters, while the grid cell has an area of ~2500 km2, possibly none of the sites will truly qualify as HOM. But if we want to keep using tower fluxes, we need to live with this. Perhaps a simple measure to reduce the effect of this tower-fetch and cell-scale mismatch is to try to work at finer resolutions in the future. GLEAM, MOD16 and Zhang, 2010 are already at resolutions 0.25 deg. PT-JPL will**

be soon available at 5 km. Perhaps a couple of products will never be available at 0.25 deg (e.g., MPI), but they could be downscaled to 0.25 deg and merged with the other products (e.g., by a nearest-neighbor interpolation if we do not want to add any extra information). The lack of success of any clustering of the sites based on vegetation, climate, etc, possibly is more indicative of the limitations of the tower flux, rather than limitation of the conceptual idea. Even if not terribly successful, a look at how the relative weights of the different products change for different clusters can be informative regarding the products performance for different conditions.

Yes, both are good suggestions for future work. As noted in our response to Reviewer 2, when globally gridded ET products allow us to derive DOLCE at higher resolutions, there are obviously many benefits. More tower data will also obviously help, especially with this second point – understanding which gridded products are better suited to different conditions.

As I mentioned before, using MPI as part of the weighted product is perfectly valid. But I think the main feature defining MPI for this study is not the fact that it is a statistical product, but more that it is a global extrapolation of the same tower fluxes used to derive the weights. As clearly stated in the paper, the tower fluxes have limitations, which will have an impact on the weighted product. Presumably, the MPI product will suffer from the same limitations than the tower fluxes, but these limitations will not become apparent when looking at differences with the tower fluxes, so it will not be reflected in the weights. An example could be the ET estimation for those biomes and moments when interception can be a relevant component of the fluxes. If the tower fluxes are not properly capturing this component, the physical methods that try to do so (e.g., GLEAM, PT-JPL, MOD16) may be penalized in the weight derivation, compared with MPI, but their fluxes may be more correct. Of course, the problem is how to merge or validate the ET products away from the tower flux data. Given that DOLCE is a global product with a relatively long time series, at least the annually integrated ET could be compared with basin-integrated differences of precipitation and runoff, which may shed more light into the merits of the individual products, the equally weighted mean product, and DOLCE.

Yes. We hope that our addition of DOLCE without the MPI product to the manuscript (as described above), and the fact that it still performs well, goes some way to alleviating this concern. At the annual scale, we could indeed neglect the change in water storage, and compare evapotranspiration with the difference (Precipitation - Runoff). This difference could serve as a test dataset for DOLCE, the equally weighted mean and the reference ensemble, away from flux data, and is independent from both MPI and DOLCE. However, this validation exercise requires observational datasets for precipitation and runoff. While there is indeed a big network for observed precipitation, gridded products can vary by surprising amounts, and currently there are no global time varying observational estimates of runoff that we are aware of. Year-to-year storage changes are also not necessarily negligible. Nevertheless, validating DOLCE as a component of the water balance is part of our next piece of research… so we clearly agree this is an aim worth pursuing.

---

## Author Comment (AC3) · 3 Jul 2017

**Manuscript hess-2017-147 entitled "Derived Optimal Linear Combination Evapotranspiration (DOLCE): a global gridded synthesis ET estimate"**

We would like to thank the reviewers for their constructive comments on our manuscript. This document outlines our point-by-point responses to the reviewer #2 comments and the improvements made to the manuscript.

**Response to Reviewer #2**

1. **I have questions about why do you choose to produce DOLCE at 0.5 deg. resolution? Not 0.25 deg or 0.05 deg (the possibilities provided by MOD16, GLEAM ET), even as high as 1km (the possibility will discussed later in my report). I need a strong rebuttal from the authors here.**

   It would be fantastic to produce a meaningful 0.05 degree global ET product, but among all the weighting products, only MOD16 has information at this scale. It could be possible to produce an ET product at 0.25 degree, but both PML and MPI have no information at this scale. One of the benefits of DOLCE is that it combines the strengths of many estimates, one of the limitations is that it requires many estimates in order to provide better information. So, given the very small number of datasets available at high-resolution DOLCE would likely not provide better information.

2. **In addition, before producing DOLCE, are the weighting ETs resampled to 0.5 degree? Or how did you combine ET at different resolution into 0.5 degree DOLCE?**

We thank the reviewer for their comment, this was indeed not mentioned in the manuscript. Yes, we resampled the 3 GLEAM products to 0.5 degree. We've included this information in the manuscript:

*The 3 GLEAM products were resampled to 0.5° using bilinear interpolation to match the spatial resolution of DOLCE*

3. **My understanding is that ET product at either 0.5, or 1 deg., most of the flux site grid is in-homogeneity, thinking about the land surface covers can varies a lot in 50km\*50km resolution. The more higher resolution of ET, the more higher chance could the site measurement represent information for a gird. Please check the reference: (Anderson et al., 2012).**

The reviewer is of course correct, there is a huge mismatch between the fetch of the flux tower (i.e. $1km^2$ or less) and the area of the grid cell containing the site, and a grid cell with an area close to the flux site fetch is more likely to be homogeneous than a 50km\*50km grid cell. Unfortunately, as we responded to point 1 above, creating DOLCE at a finer resolution was not possible. We also dedicated a considerable portion of the manuscript to exploring the representativeness question (as noted by Reviewer 1), by testing whether the weighted DOLCE product performed well at sites that were not used to derive the weights (the results shown in Figures 3 and 4, and in particular the two columns of plots in Figure 4). We have also added text to make it clear that DOLCE will *not* perform better at every site, but only perform better across a wide range of sites considered collectively:

*In both cases it is important to note that many individual sites agree poorly with DOLCE compared to some other products. The distinction between the results shown in Fig. 3 versus Fig. 4 serve to highlight that DOLCE, and indeed any other large scale gridded ET product, is not suitable for estimation of an individual site's fluxes, even if prediction over many sites shows notable improvement.*

**4. DOLCE in this work does not make an extension for spatial resolution, or temporal resolution, even time coverage (10 years, 2000-2009). It is not encouraged for publication, which is similar to already published by Mueller et al. 2015. However, the method used to merge ET products is very useful. But the way of using it at 0.5 deg is not an optimal one.**

As discussed in point 1 above, with currently available ET products, it is not possible to produce DOLCE at a higher resolution, however it is always possible to expand the time coverage of DOLCE as its component products evolve, indeed this is the aim for future versions of DOLCE. We added a statement to this effect in the conclusion.

*Expanding DOLCE over longer time periods and incorporating more diagnostic ET datasets (such as PT-JPL, CSIRO-global, GLEAM-V3B, SEBS, WECANN and HOLAPS) will be carried out in future versions.*

**5. I would say whether the site is reported homogeneous in the evaluations is related to the scale. To asses site homogeneity by ET producer is not fare for the operation of these flux sites. I understand site PIs would seek to ensure the site represent one typical land surface. All the flux tower can be taken as homogeneous sites when the evaluated ET is at 10 meter resolution. Move to 0.5 deg grid, all the flux is located in an in-homogeneous grid. Thus you cannot say 'homogeneous sites', but 'homogeneous site grid'. Or most likely homogeneous due to good matching between 0.5 deg grid and flux site.**

This is a good point that we do need to clarify, here we use HOM and HET to describe whether the land cover attributed to the grid box (based on the IGBP map at 0.5 grid) matches the land cover at the flux tower. We clarified this in the manuscript and replaced HOM site and HET site with "HOM-case" and "HET-case" respectively.

6. **This also intrigue my interests, if you use the method to combine and calibrate existing ET to be a ET at 1km or higher resolution, which makes all available flux sites homogenous at this scale and can be used to calibrate weighting ET, more land covers can also be used not only DOLCE Tier 1, 2 and 3, say as IGBP 18 land covers. Why don't you use IGBP land covers to replace tier 1, 2 and 3 to derive weight for the 18 land covers? More classification or tiers can also help you produce DOLCE at higher resolution. You may say weighting ET cannot provide information at lower resolution.**

We thank the reviewer for their suggestion. The difference between Tier1, Tier2 and Tier3 is the size of the ensemble of ET products: Tier1 is derived from 6 ET products, Tier2 from 5 and Tier3 from 2 products. We produced Tier2 and Tier3 which have less products than Tier1 to overcome the limitation of the spatial coverage in the excluded products and ensure a global coverage of DOLCE (i.e. composite of Tier1, Tier2 and Tier3), as explained in Section 2.1.

The idea of producing a weight for each land cover is a great idea, but as explained in the manuscript we didn't have flux tower covering all biome types, and for some we had very few sites, not enough to calibrate the weights without over-fitting becoming an issue. We did try to group similar biome types so that each group maintained enough members to allow the in- and out-of-sample testing approach – this was outlined in the discussion with results presented in supplementary material. The results showed that clustering by biome type doesn't improve the weighting (figure S2).

7. **But the weighting ET can be calibrated with flux site at higher spatial resolution. Each weighting ET can be resampled and calibrated to e.g. 1km resolution. Then you can further combine them into a DOLCE high resolution ET.**

This point is clearly very similar to the first point raised above. Calibrating low resolution products with site observations to produce high resolution product might seem to be a great idea, however the vast majority of the products weighted here do not have information at fine scales, so while there would be data at high resolution, there would be no more information than the low-resolution

product. This might lead a user of the product to believe it provided more information than it actually does.

8. **This work will make the DOLCE more useful. The current 0.5 deg. ET does not differentiate from other fused/merged ET at coarse resolution, e.g. Landflux.**
We thank the reviewer for their suggestion. The out-of-sample tests show that DOLCE @0.5degree) is performing better than Landflux (@1degree) in all the diagnostics we xamined.

9. **In addition, I also think about if your collection of flux tower data is not enough, which may lead to an biased weight for DOLCE calculation. One reviewer also pointed out the limitation of flux tower for the tropical region. Especially, the flux tower play an important role in your method.**
We have addressed this point in detail in our response to point 1 raised by Reviewer 1.

10. **You mentioned that 'irrigated sites' was excluded. What's the purpose of doing this?**
We added the text below in the manuscript to clarify this point:

*We expect that some of the weighting models will largely underestimate the flux at irrigated sites, a result of a missing irrigation module in their scheme (Jung et al., 2011; Miralles et al., 2011). Because of this, the error bias of these models at the irrigated sites will modify the mean error bias (i.e. mean bias across all the sites) significantly, which will affect the weighting in favour of the products that can represent better irrigation. We excluded these sites as we do not want the products to be weighted for their inclusion/non-inclusion of physical processes.*

11. **is it believed that all the weighting ET product cannot estimate ET for irrigated crops? If yes, this means DOLCE can also has a big errors for irrigated regions or human influenced regions. Then this need pointed out or at lease add discussion about shortage of this dataset. If no, I would be interested to look at the performance of weighting ET products and**

**DOLCE at irrigated flux sites, since irrigated crops may also influence global water balance. We cannot blind to this issues when producing a global ET. Can't we? I agree if you remove irrigated sites or HT sites, this will makes your ET or paper looks better, but in reality it also expose the shortage of the method and products.**

Testing how DOLCE performs at irrigated sites is a good idea. We expanded our analysis to evaluate how DOLCE performs at the three excluded irrigation sites. We displayed the results in Table 2 and added the text below in the Result section:

*We tested the performance of DOLCE at three irrigated sites that were excluded from the weighting for reasons explained earlier (section 2.2), by computing the four statistics. A description of these sites and the results are shown in Table 2. The results show that the performance of DOLCE is reasonable at US-Ne1 and US-Ne2 and low at US-Twt. These results are discussed further below.*

We also discussed the results by including this text in the Discussion section:

*DOLCE has also shown a weak performance at US-Twt, which is an irrigated rice paddy, this site gets flooded in spring and drains in early fall, then the rice is harvested. Only 9 months were available for this site, which coincide with the flood and drain period between spring and fall. DOLCE couldn't depict the flooding and draining event, probably because none of the weighting products can represent such phenomena, so it is expected that representing seasonal flooding is a shortage in DOLCE.*

**More specific comments list below.**

**12. There are a lot of errors in the Table 1. This has been pointed out by Carlos Jimenez.**

We thank the reviewer for spotting this, we have corrected these.

**13. In addition, I also found some ET not included. This is also a 0.05 deg. monthly ET. You may find here: http://en.tpedatabase.cn/portal/MetaDataInfo.jsp?MetaDataId=249454**

A great suggestion. Yes, SEBS is a remote sensing product and looks a good addition to the weighting ensemble. In the current version of DOLCE, we considered global ET products that cover the whole period 2000-2009, whereas SEBS misses a few months in 2000. In the future versions of DOLCE, this will not be an issue and we will consider adding SEBS to the weighting ensemble.

**14. Table3. Weighting ET products have negative or positive mean bias, if you give 0.5 weight to MPIBGC (3.837 mean bias) and GLEAM-v2B (-3.571) respectively, then DOLCE tier 1 will have a mean bias of 0.266 W/mˆ2. I also calculate mean bias with the weight and mean bias in table 3 for tier 1, then I get a mean bias of 0.9021 (0.041*3.756+0.495*3.837+(-0.026*6.180+....=0.9021), why do you think the weight provided in table 3 is better than my suggested weight of the two ETs?**

The bias represents the mean error of each product with respect to the observation. In the weighting technique, we bias correct the product first, then apply the weighting. This was perhaps not clear enough in the manuscript, so we've changed the table caption to read:

*Table 3: (1) Bias of weighting products and (2) weights assigned to the bias corrected products in the case of each of the three DOLCE tiers, and the number of flux tower sites used to feed the weighting.*

So, to answer this question (a) we are not optimising for mean bias, and (b) we are testing out of sample to make sure the weighting approach works well.

**15. Page 11 line28-30, please see my comments above**

Thanks, this point has been discussed above.

**16. Page 5, at what time resolution did you do the energy balance correction in the two equations or methods? Monthly or half-hourly? I would not expect 'no qualitative differ-ences'**

**between bowen and residual term correction at half-hourly flux data. Secondly, most of the flux sites have no ground heat flux but soil heat flux at 5cm.**

We added the text below in the manuscript to clarify this point:

*We used daily averages of latent heat flux represented by "LE_CORR" in FN dataset. In LT dataset, we employed the components of energy imbalance and their associated flags (in brackets), represented by G_f (G_fqcOK), for soil heat flux, H_f (HFqcOK) for sensible heat flux, Rn_f (Rn_fqcOK) for surface net radiation and LE_f (LE_fqcOK) for latent heat flux.*

**17. What's your consideration of G used in the correction.**

We applied quality control and filtering for G as highlighted in steps (2) and (3), section 2.2. We added the text below to clarify this point:

*Applying a correction technique for energy imbalance at LaThuile sites required applying (2) and (3) for the other components of energy imbalance (i.e. $R_n$, G and H), which means that the sites that had to undergo a correction for the energy imbalance, should have monthly estimates for all the fluxes of the energy budgets, where each monthly value has been calculated from at least 15 daily mean flux values. Because of this constraint, many sites were disregarded from the analysis.*

**18. Fig.4 LFD or LDF*,LFA or LFA*?**

The reviewer is right, we now changed LDF label to LFD and we modified the caption to clarify why we added * to the four products. The modified caption of Fig.4 reads:

*Figure 4: In (a), (b) and (c), as for Fig. 3 but showing the one site out-of-sample tests. Box and whisker plots are generated through selecting one site to be out sample and are repeated for all 138 sites. Products marked with * have limited spatiotemporal availability relative to the diagnostic ensemble, and testing against the LFA, LFD, CS and PT products was limited to 110, 108, 108 and 72 sites respectively. In (d), (e) and (f), the one out-of-sample test is trained by HOM-case sites data only.*

**19. Figure 7, there is no values for the Sahel desert. This is due to non-values from DOLCE or LandFlux? Then I found DOLCE has ET estimate for Sahel desert in Fig. 8. Please explain it.**

We created a mask from the spatial intersection of DOLCE, MPI and Landflux so that when we perform the comparison of DOLCE with MPI (fig. 6) and LandFlux (fig. 7) we look at the same areas. This eliminated the Sahel desert from the comparison, as it is not covered by MPI.

**20. Fig. 8. How did you say reliability of uncertainty is low? DOLCE has uncertainty with monthly temporal resolution, am I right? I have difficulty understanding 'seasonal variability of global mean ET and its associated uncertainty'. It's better say 'spatial distribution of a) global ET and (b) its associated uncertainty (standard deviation)in Winter and Summer,'.**

We've added two extra plots that show the seasonal variability of 1) ET estimates and 2) uncertainty estimates, we changed the plot titles and rewrote the caption to read:

*Figure 8: Seasonal (a) global mean ET and (b) its variability (standard deviation), (c) time average of uncertainty (the standard deviation uncertainty shown in Equation 7) (d) standard deviation of uncertainty over time (e) reliability, defined as high ($\frac{Uncertainty\ SD}{mean\ ET} \leq 1$ in blue), medium($|mean\ ET| \leq 5$, $Uncertainty\ SD < 10$ and $\frac{Uncertainty\ SD}{mean\ ET} \geq 1$ in green) and low (in red). DJF is shown in the left column and JJA in the right column.*

**21. Line 10. 'Together with the reasonable density .. are reasonably well constrained.' need rephrase.**

We have rewritten this to read:

*These datasets together with in-situ surface observations have provided constraint on the reanalysis products that provide the basis of global gridded LSM forcing products.*

**22. Page 1, Line 16, 'point-based estimates of flux towers provide information at the grid scale of these products.', are you sure point flux tower can provide information at 50*50 km pixel? Please check my comments above.**

As noted above in our response to point 3, we also think this is an interesting question that we have gone to some lengths to address – please see our response above for more detail. The fact that overall, the weighting is succeeding out-of-sample shows that the point-based measurements used to weight DOLCE do indeed provide useful information at the 0.5 degree grid cell scale. So yes, within the caveats noted in the revised manuscript, we are reasonably certain that this is the case.

**23. Page 1, Line 21, These ET products differ in their data requirements, the approaches used to derive them and their estimates (Wang and Dickinson, 2012). This is well known. No need citation here.**

We acknowledge that many in the community do know this, but are reasonably certain that others do not, and so have left the reference as is – there is no cost to this.

**24. Line 24, we provide an even stronger vindication of this relationship. Which part show this? Please give some explanation.**

In section 2.3 we showed that combining products will always derive a better performing product at the in-sample sites. We now refer the reader to Section 2.3.

**25. Please use either 'Time-space step' or 'Space-time step';**

We thank the reviewer for spotting this. We now made the necessary changes and chose to use "Time-space".

**26. Acknowledgements Where did you use ERA-inerim by saying 'The ERA-Interim reanalysis data are provided by ECMWF and processed by LSCE'.**

The reviewer is right, we haven't used ERA-Interim reanalysis data, but part of the term and conditions of the use of FLUXNET data is to use the exact statement included in http://fluxnet.fluxdata.org/data/data-policy/. We assume ERA-Interim is included in the statement because it was used for gap filling.

27. **References: Annan,.....n/a-n/ Fisher, JB has been listed two times. Miralles, D.G, also listed two times. Please check the standard format for references used on HESS website**

Thanks for picking this up, we have removed the duplicates and made the appropriate corrections.

10 28. **Table s1, please also add RMSE, correlation and mean bias values for each site. This information is also important for DOLCE dataset users.**

We obviously have all these values and they are all included in the plots (Figure 5 and Figure 9). The table is already dense but we are very happy to include them if the Editor thinks that fitting these into the table will not make the table unpublishable.

---

## Author Comment (AC4) · 3 Jul 2017

The comment was uploaded in the form of a supplement:
https://www.hydrol-earth-syst-sci-discuss.net/hess-2017-147/hess-2017-147-AC4-supplement.pdf

---

## Author Comment (AC6) · 3 Jul 2017

The comment was uploaded in the form of a supplement:
https://www.hydrol-earth-syst-sci-discuss.net/hess-2017-147/hess-2017-147-AC6-supplement.pdf

---

## Referee Report (RR1)

To assess 0.5 deg. ET product with flux towers is a kind of silly problem. Why creating ET at a finer resolution (0.05) was not possible, by using the ensemble weighting and rescaling technique? Indeed, if there is a way for higher resolution, I suggest to rethink on this. As in next years, fine resolution ET will come out. In addition there are other ET dataset, such as PTJPL, SEBS, GLDAS, ERA-Interim, which should be used to enlarge your ET input source, if you contact the groups. If the author doesn`t expand the time coverage or spatial resolution before the paper is published, it is mostly likely not possible for DOLCE ET updated after the publication. As bother reviewer and editor have suggested using other global ET, but no real reaction is adopted. Don`t agree that there is not enough flux towers for each land cover and biome types. Even the clustering by biome type doesn`t improve the weighting, however, it can help you derive a high resolution ET. Answers to comment 7, whether to do spatial interpolate or not does not influence your flux tower evaluation result? Please re-check this. If I am right, here you are selecting the pixel value where the flux tower be located in to match with flux observation. However, you can also use 2-d spatial interpolation to get the point ET value with the geo-location of the tower. Please check if this will influence your weight, mean bias, and SD. Then you can say it`s not necessary to calibrate weighting ET at higher spatial resolution with flux observations. Please remember you are not satisfying the reviewer but the potential DOLCE ET users.

---

## Author Response (AR2)

**Manuscript hess-2017-147 entitled "Derived Optimal Linear Combination Evapotranspiration (DOLCE): a global gridded synthesis ET estimate"**

Dear Dr Su,

Thank you for your response and taking the time to seek additional reviews. Below we discuss both of the points you raise, as well as the additional points raised by the reviewer, and some additional investigation we did to allay these concerns. We start by attempting to give a clearer justification for the position we have taken on these in the manuscript, before outlining possible ways forward.

**Reviewer #2 pointed out that it is not convincing to compare 0.5 degree grid data to in-situ flux tower measurement. I would suggest that you consider a scaling technique based on land cover so that the two are comparable. A simple foot-print consideration should also be very useful.**

We wholeheartedly agree that point-scale tower measurements, with a footprint no larger than 1km$^2$, cannot be assumed to be representative of each 0.5 degree grid cell. However we feel that the reviewer's assertion "to assess 0.5 deg. ET product with flux towers is a kind of silly problem" misrepresents the fact that we explicitly addressed this issue in great detail and also does not acknowledge that almost all existing gridded ET products have been tested in this way in existing published research (e.g. McCabe et al., 2016).

We dedicated a significant part of the introduction to establish that this is a problem with existing published approaches and developed an experimental approach to explicitly test the validity of this assumption. Indeed we feel it is one of the most novel aspects of what is presented here, since almost all existing studies effectively ignore this issue.

We have made changes to the manuscript to try to make this clearer, which we will outline below, but first restate our approach as it was submitted to demonstrate this issue is far from ignored. In the introduction we note that:

*"In each of the evaluation studies described above, tower data from FLUXNET provide ground truth for gridded ET datasets by comparing grid cell values to those measured at the site scale. Most gridded ET products have a 0.5-degree resolution, so that each grid cell can represent an area of around 2500 km$^2$. The fetch of flux tower measurements varies depending on terrain, vegetation and weather, but is typically under 1 km$^2$ (Burba and Anderson, 2010). None of these studies directly address this obvious scale mismatch, and the degree to which surface heterogeneity might nullify any information that flux towers provide about fluxes at these larger scales."*

*"We examine the performance of the weighting approach in several in-sample and out-of-sample tests that confirm flux towers do indeed provide information at the grid scale of these products."*

In Section 2.4 we detail how we can test whether point scale measurements do, on average, actually provide any information about the 0.5 degree scale. This is done by testing whether the weighted combination of existing products, that clearly will improve results against site data with which it is trained, can still deliver improvements at sites that were not included in its training data set. If it does better than no weighting at these unseen sites, then there must be information content about the larger scales in the site data. This is precisely what is shown in Figures 3, 4, 11 and 12 – all of these performance results are ONLY for sites that were not used to train the weights (i.e. they are out-of-sample tests).

While the results are not spectacularly good, they are solid:

*"Critically, the fact that the weighting improves out of sample performance suggests that while the representativeness of point-scale measurement for the grid scale may not exist at every single site, it does exist across all these sites as a whole."*

We then showed that restricting the application to sites where the tower was more representative of the 0.5 degree grid cell did indeed improve results further. We also feel that we are clear about caveats in this relationship:

*"The distinction between the results shown in Fig. 3 versus Fig. 4 serves to highlight that DOLCE, and indeed any other large scale gridded ET product, is not suitable for estimation of an individual site's fluxes, even if prediction over many sites shows notable improvement."*

And feel that our conclusion of this part of the investigation clearly reflected the findings:

*"It was shown that despite the scale mismatch between the flux tower and the grid cell, the ensemble of flux towers as a whole can provide information about the grid cells that contain them. While the representativeness of the point scale for the grid scale is enhanced by only considering sites that lie within homogeneous grid cells we suggest that an optimal definition of homogeneity for flux behaviour be the subject of future investigation."*

In short, we feel that the reviewer's one sentence dismissal is an unfair portrayal of what the manuscript contains. We note that the other reviewer, Paul Dirmeyer, explicitly endorsed the approach we took regarding this issue.

Nevertheless, we have clarified this further in the results section :

*" It is important to reinforce that these results are for sites that were not used to train the weights. As detailed in section 2.3, performance improvement at training sites is expected, but the fact that the weighting delivers improvements at sites that were not included in training data indicates that there is indeed information content about the larger scales in site data.."*

And in the conclusion:

*"It was shown that despite the scale mismatch between the flux tower and the grid cell, the ensemble of flux towers as a whole does provide information about the grid cells that contain them, since the improvements delivered by the weighting approach were evident in sites not used to derive the weights."*

**2. Another issue is your use of available ET data - I am in the same opinion as reviewer #2 that you should consider more available data. I do not think your response to this concern is convincing as such.**

We are absolutely interested in using as many products as possible, and can incorporate more relatively easily. Our primary aim for DOLCE is land surface model evaluation, and as such, we have avoided including any products that are derived using similar model structures, such as reanalyses.

We also thought of including other products, in particular SEBS, CSIRO-global and PTJPL, however none of them have a full coverage of the period 2000-2009. This period was in fact chosen to maximise the number of products we could include while still having a final product that was at least a decade in length.

One potential way to address your concern is to derive DOLCE with different component products in different time periods, however we feared that doing so would lead to temporal discontinuities in the derived product. We have now stated as much in section 2.1:

> *" The reasons for restricting the Diagnostic Ensemble are (a) to maximize the time period covered by DOLCE (see Table 1), (b) to avoid temporal discontinuities in the derived product that can result from using different component products in different time periods ,(c) to maximize the number of flux tower sites that can inform the weights (noting that datasets have different spatial coverage), and (d) to avoid LSM-based estimates in the final DOLCE product, so that its validity for LSM evaluation is clearer."*

If you have a suggestion for how this could be made clearer please let us know.

**Response to Reviewer:**

We also address the reviewer's concerns individually here:

> 1. *To assess 0.5 deg. ET product with flux towers is a kind of silly problem.*

Please see our detailed response above – in short, we feel that this comment ignores the very extensive treatment of this issue in the manuscript, particularly that we clearly showed as part of the results that this was not a 'silly problem', that flux towers did in fact reliably provide information at the 0.5 degree scale. We also feel that the comment ignores that most constituent products used here were evaluated, and published, in this way.

> 2. *Why creating ET at a finer resolution (0.05) was not possible, by using the ensemble weighting and rescaling technique? Indeed, if there is a way for higher resolution, I suggest to rethink on this. As in next years, fine resolution ET will come out.*

As noted in our previous response to this same point, this is not the aim of our study, and only one product has information at this resolution, so even if we wished to apply the novel methodology in this paper to this scale, it would simply not be possible.

> 3. *In addition there are other ET dataset, such as PTJPL, SEBS, GLDAS, ERA-Interim, which should be used to enlarge your ET input source, if you contact the groups.*

As explained in more detail above, we would love to have more products in DOLCE, indeed our approach to component product selection was precisely to maximise the number of products we could include, but for the given time window, and our desire to keep the product applicable to land surface model evaluation, this is the maximum set currently available to contribute.

4. *If the author doesn`t expand the time coverage or spatial resolution before the paper is published, it is mostly likely not possible for DOLCE ET updated after the publication.*

This seems an odd comment. Why would a change in time coverage or spatial resolution affect our ability to update the product in the future? At least to us, this is incorrect.

5. *As bother reviewer and editor have suggested using other global ET, but no real reaction is adopted.*

We assume this is the same concern raised in point 3 above – please see our response to (3) and to the editorial comment above.

6. *Don`t agree that there is not enough flux towers for each land cover and biome types.*

This is simply about statistics. While we can fit several ET products to a small amount of flux tower data, this will result in overfitting and poor out of sample performance. It is an extremely widespread rule of thumb in statistical literature that the bare minimum number of data points required for a meaningful regression relationship is approximately ten times the number of predictors. We can provide references if needed, but feel this is fairly rudimentary.

7. *Even the clustering by biome type doesn`t improve the weighting, however, it can help you derive a high resolution ET.*

Please see our response above, and in our first set of responses as to why we do not derive a higher resolution product.

8. *Answers to comment 7, whether to do spatial interpolate or not does not influence your flux tower evaluation result? Please re-check this. If I am right, here you are selecting the pixel value where the flux tower be located in to match with flux observation. However, you can also use 2-d spatial interpolation to get the point ET value with the geo- location of the tower. Please check if this will influence your weight, mean bias, and SD. Then you can say it`s not necessary to calibrate weighting ET at higher spatial resolution with flux observations.*

Firstly, as explained in our response to (2) above, we note that a higher spatial resolution product using the novel methodology in this paper is simply not possible, as only one component product contains information at an appropriate scale. It is also not the aim of this study.

Spatial interpolation at 0.5 degree to compare tower and grid cell data is of course possible, and is indeed a sensible suggestion. To understand whether this was likely to have any qualitative effect on our results, we compared the grid cell values of DOLCE that contained each flux tower with DOLCE values at the towers using bilinear interpolation. Results are shown below. The first panel compares the ET values of grid cells ('DOLCE') with

interpolated values ('DOLCE_i'), as well as box and whisker plots of the differences between these two. The remaining panels show comparisons to flux tower data using the key metrics used in the paper – as is shown in Figure 5 of the manuscript. As you can see this makes no qualitative difference to the results. As such we see no reason to further complicate the methodology of the paper, but have noted that this avenue was explored in the discussion section of the manuscript:

> *"We also investigated using bilinear interpolation instead of direct grid cell to tower comparison (not shown), but found no qualitative differences."*